# The out-of-field dose in radiation therapy induces delayed tumorigenesis by senescence evasion

Erwan Goy[1], Maxime Tomezak[1,2], Caterina Facchin[1], Nathalie Martin[1], Emmanuel Bouchaert[3,4], Jerome Benoit[3,4], Clementine de Schutter[1], Joe Nassour[1], Laure Saas[1], Claire Drullion[1], Priscille M Brodin[5], Alexandre Vandeputte[5], Olivier Molendi-Coste[6], Laurent Pineau[6], Gautier Goormachtigh[1], Olivier Pluquet[1], Albin Pourtier[1], Fabrizio Cleri[2], Eric Lartigau[7], Nicolas Penel[7], Corinne Abbadie[1]*

[1]Univ. Lille, CNRS, Inserm, CHU Lille, Institut Pasteur de Lille, UMR9020-U1277 - CANTHER - Cancer Heterogeneity, Plasticity and Resistance to Therapies, F-59000 Lille, France; [2]Univ. Lille, CNRS, UMR8520, Institut d'Electronique, Microélectronique et Nanotechnologie, F-59652 Villeneuve d'Ascq, France; [3]Oncovet Clinical Research, Plateforme PRECI, F-59120 Loos, France; [4]Oncovet, Plateforme PRECI, F-59650 Villeneuve d'Ascq, France; [5]Univ. Lille, CNRS, Inserm, CHU Lille, Institut Pasteur de Lille, U1019 - CIIL - Centre d'Infection et d'Immunité de Lille, F-59000 Lille, France; [6]Univ. Lille, Inserm, CHU Lille, Institut Pasteur de Lille, U1011 - EGID, F-59000 Lille, France; [7]Lille University, Medical School and Centre Oscar Lambret, Lille, France

*For correspondence:
corinne.abbadie@ibl.cnrs.fr

Competing interest: The authors declare that no competing interests exist.

**Abstract** A rare but severe complication of curative-intent radiation therapy is the induction of second primary cancers. These cancers preferentially develop not inside the planning target volume (PTV) but around, over several centimeters, after a latency period of 1–40 years. We show here that normal human or mouse dermal fibroblasts submitted to the out-of-field dose scattering at the margin of a PTV receiving a mimicked patient's treatment do not die but enter in a long-lived senescent state resulting from the accumulation of unrepaired DNA single-strand breaks, in the almost absence of double-strand breaks. Importantly, a few of these senescent cells systematically and spontaneously escape from the cell cycle arrest after a while to generate daughter cells harboring mutations and invasive capacities. These findings highlight single-strand break-induced senescence as the mechanism of second primary cancer initiation, with clinically relevant spatiotemporal specificities. Senescence being pharmacologically targetable, they open the avenue for second primary cancer prevention.

## Editor's evaluation

A major issue from radiation therapy is the generation of secondary tumors that can arise a long time after treatment. Here, the authors use careful and innovative experimental systems to show that out-of-field dose scattered radiation induces a senescent arrest characterized by single-strand DNA breaks (SSB) and the ability to escape from the senescent arrest albeit at a very low frequency. The data are consistent with a key role for SSBs and the ability of the cells to escape from senescence, and the paper has a clear clinical potential: the prevention of secondary tumors using senolytic agents.

## Introduction

Curative-intent radiation therapy relies mainly on the generation of DNA damages lethal for cancer cells. Paradoxically, because DNA damages are mutagenic, ionizing radiations also have tumorigenic effects. A severe complication of radiation therapy is therefore the induction of second primary cancers (SPCs), also called radiotherapy-induced second cancers, second cancers after radiation therapy, or cancers in irradiated fields (*Berrington de Gonzalez et al., 2013*; *Doyen et al., 2010*; *Travis, 2006*; *Trott, 2017*; *Trott, 2017*; *Tubiana, 2009*; *Vautravers et al., 2010*). SPCs are not a recurrence of the initial cancer but are neo-formed from normal cells affected by the radiations. Data on the increased risk of developing an SPC after radiation therapy is quite divergent in the available studies: it ranges from 0 in a cohort of 13,457 patients with rectal cancer having received radiation therapy after surgery compared to surgery alone (*Martling et al., 2017*) to 14 in a cohort of 1136 patients diagnosed with Hodgkin lymphoma at a median age of 11 years (*Holmqvist et al., 2019*).

Although the causal relationship between the initial radiation therapy and the induction of the SPC is impossible to definitely prove, SPCs are highly suspected to be radiation-induced. However, lifestyle, exogenous stressors, and genetic factors could also be contributing factors (*Doyen et al., 2010*). Amazingly, in radiation therapy using external X-rays, SPCs develop not preferentially inside the planning target volume (PTV), that is, the volume having received 100% of the therapeutic dose, but preferentially close to the edge of the beams, or farther around, on several centimeters (*Diallo et al., 2009*; *Dörr and Herrmann, 2002*). In a clinical linear accelerator, the region surrounding a beam (which will be called the margin in this article) receives photons leaking and scattering from the primary and secondary collimators, scattering from the flattening filter and scattering inside the patient from the PTV toward normal peritumoral tissues (*Chofor et al., 2012*). Although the marginal dose (also called peripheral dose or out-of-field dose) emanating from the collimator and the flattening filter could be reduced by optimizing the accelerator's technology, the scattering inside the patient is unavoidable. This internal scatter component was shown to be the major determinant of the dose deposited in the most proximal margin (*Chofor et al., 2012*). The marginal radiation has three characteristics: (i) the deposited dose exponentially decreases with the distance, (ii) the deposited dose is approximately proportional to the PTV size, and (iii) the photon's spectral energy fluence distribution is shifted to lower energies compared with those inside the PTV (*Chofor et al., 2012*; *Chofor et al., 2011*; *Kirkby et al., 2007*; *Kry et al., 2007*; *Wiezorek et al., 2009*). It was suggested that this photon quality change has the potential to increase the relative biological effectiveness (RBE) (*Chofor et al., 2011*; *Kirkby et al., 2007*), but the underlying specific cellular damages, if any, are unknown.

X-rays induce, among others, DNA single-strand breaks (SSBs) and double-strand breaks (DSBs) (*Azzam et al., 2012*; *Jonathan et al., 1999*; *Panganiban et al., 2013*). DSBs are defined as two breaks in the sugar-phosphate backbone on each DNA strand, at less than 10 bp. Their signalization and repair involve the activation of the DNA damage response (DDR) pathway with, downstream, the activation of the tumor suppressor TP53 that induces either a transitory cell cycle arrest favoring DNA repair or apoptosis. An SSB is a break in the sugar-phosphate backbone on only one DNA strand, often accompanied by a nucleotide loss and often displaying abnormal 5' and 3' ends (*Caldecott, 2008*). The SSB repair (SSBR) pathway involves a first step of break detection initiated by the poly(ADP)ribose polymerase (PARP) 1, which, once activated by its linkage to the broken DNA, synthetizes long chains of poly(ADP)ribose (PAR). The accumulated PAR chains favor the recruitment of the X Ray Repair Cross-Complementing Group (XRCC) 1 scaffold protein. XRCC1 is then phosphorylated and recruits the repair enzymes themselves, including the polynucleotide kinase phosphatase (PNKP) and the polymerase β. These last steps are common with the base exchange pathway (BER) (*Caldecott, 2014*). It is known for a long time that the curative effect of ionizing radiations results from the formation of DSBs too many to be repaired, leading to cancer cell death by apoptosis or mitotic catastrophe (*Jonathan et al., 1999*; *Norbury and Hickson, 2001*; *Panganiban et al., 2013*). SSBs are less detrimental than DSBs, but 1 Gy deposited in the PTV generates 10–25 more SSBs than DSBs (*Dahm-Daphi et al., 2000*). Presently, nothing is known about the respective quantities of SSBs and DSBs generated in the margin of the PTV and the consequences on the outcome of the affected normal cells.

Besides their spatial distribution related to the PTV, another amazing property of SPCs is their long latency period before emergence, ranging from 1 to 40 years after the treatment of the first cancer (*Berrington de Gonzalez et al., 2013*; *Dörr and Herrmann, 2002*; *Doyen et al., 2010*; *Sheppard and Libshitz, 2001*; *Tubiana, 2009*; *Vautravers et al., 2010*). Moreover, the cumulative incidence

of SPCs increases with time (*Doyen et al., 2010*; *Friedman et al., 2010*; *Olsen et al., 2009*; *Storm, 1988*): for instance, the cumulative incidence of second sarcoma in a cohort of 16,705 patients treated by radiation therapy for nonmetastatic breast carcinoma was calculated at 0.7% at 5 years, at 0.27% at 10 years, and at 0.48% at 15 years after the initial treatment time (*Kirova et al., 2005*). This suggests that the normal cells that were affected by the scattered photons remain dormant in the organism while maintaining a constant neoplastic transformation potential. However, the precise biological form of this dormant cell state is completely unknown. We assayed in this study whether it was senescence.

Indeed, although initially described as the state reached by normal human fibroblasts after a finite doubling number, senescence is now recognized as a more general reprogrammed state established to adapt to several stresses, such as telomere shortening, DNA damages, oxidative damages, or hyperactivation of some oncogenes (*Abbadie et al., 2017*; *Ben-Porath and Weinberg, 2005*). By producing DNA damages and oxidative stress, ionizing radiations are senescence inducers (*Cmielová et al., 2011*; *Panganiban et al., 2013*; *Seol et al., 2012*; *Suzuki et al., 2003*). The senescence program includes an increase in cell size, complex epigenetic changes, an increase in reticulum endoplasmic stress, an increase in autophagy, changes in the composition of the secretome, and above all a long-term cell cycle arrest in G1 associated with a resistance to apoptosis, making senescent cells nonproliferating long-lived cells in vivo (*Abbadie et al., 2017*; *Martien and Abbadie, 2007*; *Pluquet et al., 2015*). The senescent cell cycle arrest results from the persistence of unrepaired DNA damages. When these damages are of a type signalized through the DDR pathway, such as DSBs or shortened telomeres, the cell cycle arrest is very stable, almost irreversible, considered as tumor suppressor (*Campisi and d'Adda di Fagagna, 2007*). In contrast, when the unrepaired damages are SSBs, as we have shown in a seminal work on endogenous oxidative stress-induced senescence, about one senescent cell on 10,000 systematically and spontaneously re-enters in cell cycle to give rise to a progeny of cells we named post-senescent neoplastic emergent (PSNE) cells. These cells display mutations, markers of transformation, markers of epithelial-to-mesenchyme transition, and are able to form small hyperplasias or carcinomas in in vivo assays (*Deruy et al., 2010*; *Gosselin et al., 2009a*; *Malaquin et al., 2013*; *Martin et al., 2014*; *Nassour et al., 2016*).

We investigated in this study whether SSB-induced senescence could be a step in the generation of SPCs after conformational 3D radiotherapy using X-rays. As SPC type, we focused on sarcomas that represent a large part of SPCs (*Diallo et al., 2009*), and whose outcome is very poor with a survival rate at 5 years of 10–40% (*Depla et al., 2014*; *Vautravers et al., 2010*). We report that normal fibroblasts located in the margin of a PTV receiving 2 Gy/day as patients enter after 2 weeks of treatment in premature senescence. This senescence induction is associated with an accumulation of SSBs in the almost absence of DSBs. We also report that about 10 days after the end of radiation fractions a few senescent cells re-enter cell cycle and generate a progeny of cells resuming proliferation and displaying mutations and increased invasive capacities. These findings lead us to propose for the first time a mechanism explaining how photons scattering inside the patient from the PTV toward the normal peritumoral tissues could induce a delayed development of sarcomas. They also pave the way for using senolytics to prevent second sarcoma development.

## Results
### Characterization of the experimental setting for cell irradiations

To determine the type and quantity of DNA breaks generated in a stromal tissue localized at the margin of a PTV and analyze their effects on the stromal cell behavior, we applied a simulated standard radiotherapy protocol to in vitro cultured normal human dermal fibroblasts (NHDFs), which are primary cells, using a linear accelerator, the Varian Primus CLINAC (*Figure 1—figure supplement 1A*), commonly used for treating patients by 3D conformal radiotherapy at the Anti-Cancer Center of Lille, France (Centre Oscar Lambret). The experimental setting was designed in the respect of the mode, energies, unitary dose, dose rate, total dose, and fractioning commonly applied to patients.

We first characterized the depth profile of dose deposition at 6 or 20 MV, the two energies used in photon mode with this accelerator. The results indicated that the maximum dose was deposited at 16–30 mm deep for 6 MV and at 25–45 mm deep for 20 MV (*Figure 1—figure supplement 1B*). Therefore, cell plates (6-, 12-, or 96-well plates) were put on a 2- or 4-cm-thick poly(methyl methacrylate)

(PMMA) plate for irradiation from below at 6 or 20 MV respectively (*Figure 1—figure supplement 2A*).

The lateral positioning laser was used to position the isocenter at the bottom of the wells, on which the cells adhere (*Figure 1—figure supplement 2A*). The light field was used to delimit a square PTV of 25 × 25 cm. Cell plates were positioned straddling the light field boundary indicating the isodose 50 in order to get at least one well column inside the PTV and several well columns spreading over the margin up to several centimeters from the limit of the beam (*Figure 1—figure supplement 2B*).

To check the accuracy of this experimental setting, cell plates were scanned and the deposited dose was calculated using the planning treatment system of the Centre Oscar Lambret. The dose profile diagram and the isodose curves indicated that, for the columns positioned inside the PTV, 100% of the dose was indeed deposited at the bottom of the wells, where the cells lay, and that the dose was almost homogeneous in the wells positioned inside the PTV. They also indicated that the dose exponentially dropped down in the wells positioned in the margin (*Figure 1—figure supplement 2C and D*). Notice that in this experimental setting, since the wells are independent and insulated from each other, the cells positioned at the margin of the PTV were not submitted to a bystander effect relying on molecules secreted by the cells located inside the PTV.

## NHDFs positioned at the margin of the PTV during a simulated radiation therapy are growth retarded but do not undergo massive death

As a first experiment to evaluate whether normal fibroblasts surrounding a PTV could suffer in some way from the irradiation of the PTV, we performed growth curves of NHDFs submitted or not to a simulated French standard radiation treatment made of successive weeks of radiation fractions of 2 Gy/day, except weekends. As a control of the curative lethal effect of such a simulated treatment, we followed in parallel the growth of MDA-MB231 cells (triple-negative breast cancer cells). As expected, both MDA-MB231 and NHDFs positioned inside the PTV underwent cell death after 3 weeks of irradiation. In contrast, NHDFs positioned in the margin did not die: those in the most distal margin continued to grow as control nonirradiated cells or underwent a slight slowdown, whereas those positioned in the most proximal margin underwent a growth arrest (*Figure 1A and B*, *Figure 1—figure supplement 3*).

Since ionizing radiations are able to induce cell death, we measured apoptotic and nonapoptotic cell death levels using Vybrant Fam Poly Caspases (Molecular Probes) and Annexin V/propidium iodide (PI) (Life Technologies) assay kits. The results indicated that MDA-MB231 began to suffer from caspase-independent cell death after four radiation fractions. In the meantime, NHDFs positioned inside the PTV also underwent cell death, but to a lower extent, both by caspase-dependent and -independent mechanisms. In contrast, NHDFs positioned at the margin, even the most proximal one, did not die at a significant level (*Figure 1C*, *Figure 1—figure supplement 4*).

## NHDFs positioned at the margin of the PTV during a simulated radiation therapy enter senescence

We then investigated whether the growth retardation of NHDFs positioned at the margin of the PTV could be due to senescence induction. Senescence was first assayed by flow cytometry according to three senescence markers: the SA-β-galactosidase (SA-β-Gal) activity using the fluorogenic $C_{12}FDG$ substrate, the cell size reflected by the forward scatter (FSC) value, and the cell granularity reflected by the side scatter (SSC value). The results indicate that the SA-β-Gal activity increased with successive radiation fractions, both in NHDFs positioned inside the PTV or at the margin, up to 7 cm. The activity reached in cells positioned inside the PTV aimed at that of replicative senescent NHDFs, whereas that reached in cells located at the margin was lower (*Figure 2A and B*, *Figure 2—figure supplement 1A*). Cell size and granularity also increased with successive radiation fractions at significant levels in cells positioned at the margin. In the PTV, cell size and granularity increased during the first week of treatment and then decreased (*Figure 2A and C*, *Figure 2—figure supplement 1B*). A careful examination of the dot plots and histograms indicated that two subpopulations appeared in the PTV after 3 weeks of irradiation: one comprising the biggest and most granular cells with a high SA-β-Gal activity, which are bona fide senescent cells, and another one comprising very granular but very small cells with a very low β-Gal activity, which probably correspond to dying cells (*Figure 2A*).

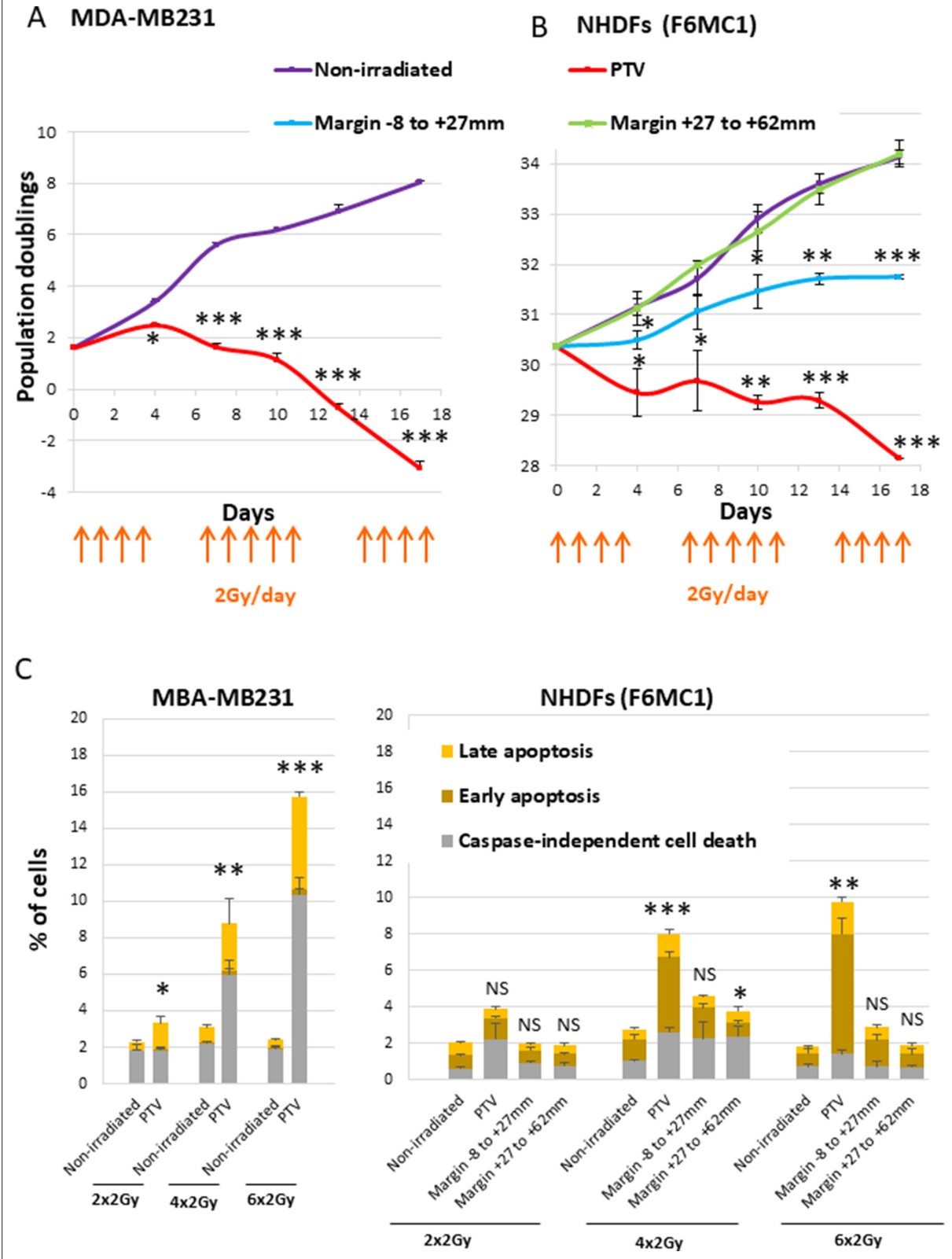

**Figure 1.** Growth curves and cell death level of MDA-MB231 and normal human dermal fibroblasts (NHDFs) positioned at the margin or inside a planning target volume (PTV) receiving 2 Gy/day. (**A**) Growth curves of MDA-MB231 positioned inside the PTV and irradiated or not during 3 weeks. (**B**) Growth curves of NHDFs (donor F6MC1, population doubling [PD] = 30.36 at the beginning of the experiment) positioned straddling the PTV and the margin irradiated or not during 3 weeks. Cells were counted at each passage using a Malassez chamber. Each point represents the mean ± SD of

*Figure 1 continued on next page*

*Figure 1 continued*

three counts from three independent culture plates. *t*-tests were performed for each counting in comparison to nonirradiated cells. (**C**) MDA-MB231 positioned inside the PTV and NHDFs (donor F6MC1, PD = 33.05 at the beginning of the experiment) positioned straddling the PTV and the margin were irradiated or not up to six times. The level of cell death was measured 8 hr after the last irradiation. Each condition was performed in triplicate. Each bar represents the mean ± SD of the three measures. Caspase-independent cell death refers to cells negative for caspase activity and positive for propidium iodide (PI). Early apoptosis refers to cells positive only for caspase activity. Late apoptosis refers to cells positive for both caspase activity and PI. *t*-tests were performed between the total percentages of cell death of each condition in comparison to nonirradiated condition.

The online version of this article includes the following source data and figure supplement(s) for figure 1:

**Source data 1.** Data for growth curves of MDA-MB231.

**Source data 2.** Data for growth curves of normal human dermal fibroblasts (NHDFs).

**Source data 3.** Data for cell death assays of normal human dermal fibroblasts (NHDFs) and MDA-MB231.

**Figure supplement 1.** Physical parameters of the linear accelerator (Varian Primus CLINAC).

**Figure supplement 1—source data 1.** Data for the depth profile.

**Figure supplement 2.** Cell irradiation setup.

**Figure supplement 3.** Growth curves of normal human dermal fibroblasts (NHDFs) from another donor positioned at the margin or inside a planning target volume (PTV) receiving 2 Gy/day.

**Figure supplement 3—source data 1.** Data for growth curves.

**Figure supplement 4.** Cell death level of normal human dermal fibroblasts (NHDFs) from two other donors positioned at the margin or inside a planning target volume (PTV) receiving 2 Gy/day.

**Figure supplement 4—source data 1.** Data for Alexa Fluor 488 Annexin Dead Cell Apoptosis assays.

**Figure supplement 4—source data 2.** Data for Annexin V-positive cells.

Another major characteristic of senescent cells is their cell cycle arrest in G1 (*Gire and Dulic, 2015*). To establish whether NHDFs also acquired this senescence marker after a simulated radiation therapy protocol, we performed a cell cycle analysis by flow cytometry. After 13 radiation fractions, NHDFs positioned at the margin were cell cycle arrested in G1, with even less cells in S phase than in a population of NHDFs at the replicative senescence plateau (*Figure 2—figure supplement 2*).

The most universally activated cyclin CDK inhibitors (CKIs) responsible for the cell cycle arrest of senescent cells are p16 (CDKN2A) and p21 (CDKN1A) (*Abbadie et al., 2017*; *Ben-Porath and Weinberg, 2005*). To establish whether the cell cycle arrest of irradiated NHDFs positioned at the margin was associated with these canonical senescence markers, we measured their expression by RT-qPCR during 2 weeks of irradiation. For NHDFs positioned inside the PTV, the results show that p16 mRNA levels increased as soon as the second radiation fraction, but then dropped back during further radiation fractions. p21 appeared to take the reins with an upregulation from the second week of irradiation. For NHDFs positioned in the margin, p16 and p21 mRNA levels followed approximately the same induction pattern, mainly from the end of the first week of irradiation (*Figure 2—figure supplement 3*). The level of p16 protein examined by Western blot and immunofluorescence followed the same expression pattern than the mRNA (*Figure 2—figure supplement 4*). These results indicate that the senescence induced in NHDFs positioned at the margin classically involved both p16 and p21.

Senescent cells are also characterized by a certain degree of multinucleation (*Dikovskaya et al., 2015*; *Matsumura, 1980*). To establish whether the cells affected by the marginal radiations could also harbor this marker of senescence, we quantified the percentage of multinucleated cells by DAPI staining followed by microscopic examination. The results indicate that the percentage of multinucleated cells approximately doubled in NHDFs positioned at the margin after 2 weeks of irradiation. This percentage did not significantly increase in NHDFs positioned in the PTV (*Figure 2—figure supplement 5*).

Taken together, these results indicate that during a simulated radiation therapy protocol, NHDFs positioned at the margin of the PTV undergo senescence, whereas those positioned inside the PTV go through a senescent-like phenotype and then die.

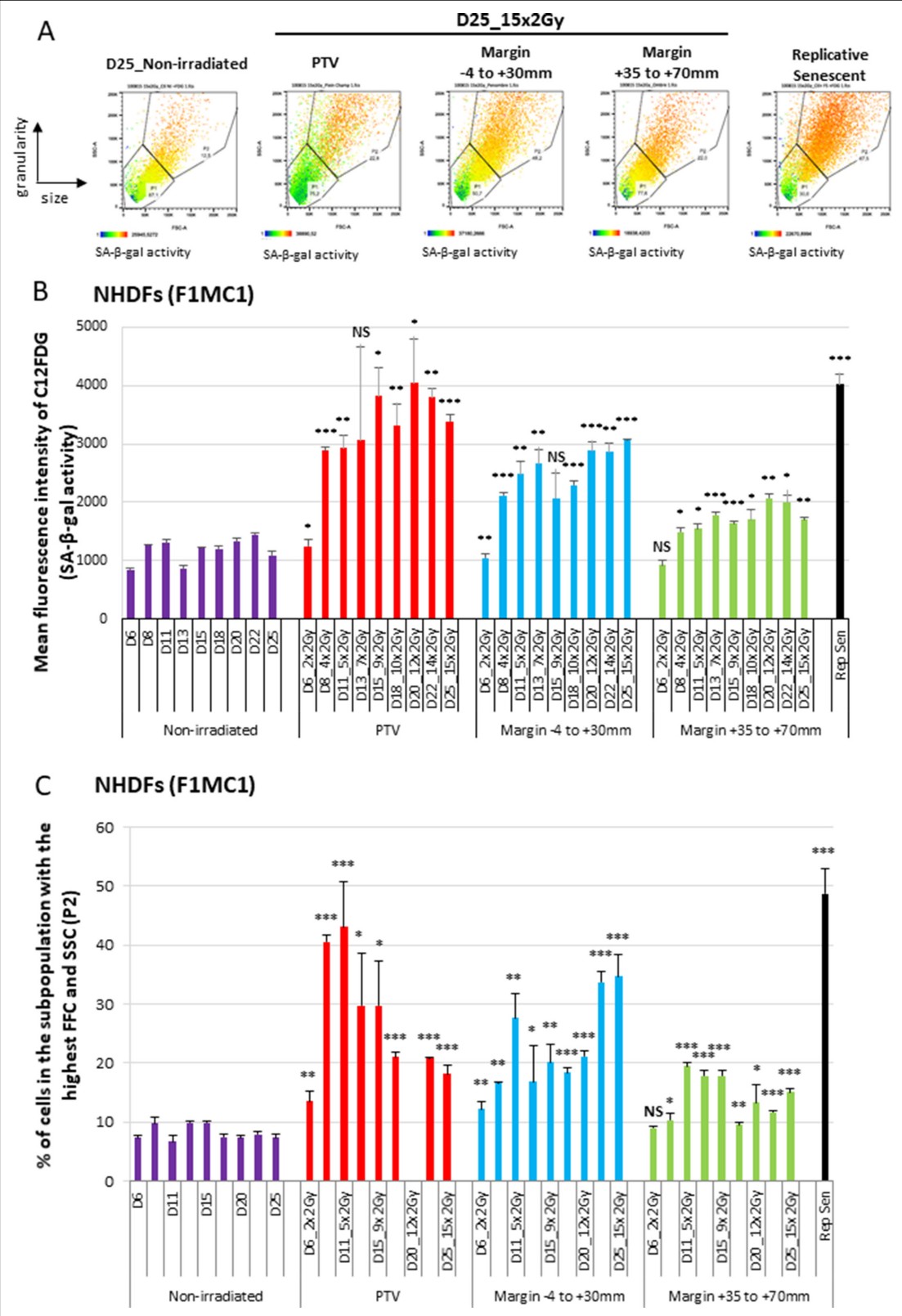

**Figure 2.** Normal human dermal fibroblasts (NHDFs) positioned at the margin of a planning target volume (PTV) receiving 2 Gy/day undergo senescence. Six-well plates of NHDFs (donor F1MC1, population doubling [PD] = 25.8 at the beginning of the experiment) positioned straddling the PTV and the margin were irradiated during 3 weeks at 2 Gy/day except weekends. Cells were analyzed 24 hr or 72 hr after the last radiation fraction by flow cytometry for their side scatter (SSC) and forward scatter (FSC) values, representing the granularity and the size of the cells, respectively, and

*Figure 2 continued on next page*

*Figure 2 continued*

for their SA-β-Gal activity using the fluorogenic $C_{12}FDG$ substrate. (**A**) Examples of how dot plots were analyzed to characterize senescent cells. The dot plots represent the size (FSC) and granularity (SSC) measures of NHDFs. The gate P2 corresponds to cells with the highest size and granularity established by comparing nonirradiated exponentially growing NHDFs to replicative senescent NHDFs. The color scale of the dots represents their SA-β-Gal activity, with the blue color representing a low activity and the orange color a high activity. The middle panels are examples of $C_{12}FDG$ measures after 15 irradiations (day 25). The right panel illustrates the gate of the biggest and most granular cells that was set by comparing nonirradiated exponentially growing NHDFs (left panel) to NHDFs at replicative senescence plateau (PD = 61.03). (**B**) SA-β-Gal activity measured 24 hr or 72 hr after the last radiation fraction by flow cytometry using the fluorogenic $C_{12}FDG$ substrate. The results are given as the mean $C_{12}FDG$ fluorescence intensity. Each condition was performed in triplicate. Each bar represents the mean ± SD of the three measures. *t*-tests were performed for each condition in comparison to nonirradiated cells. (**C**) SSC and FSC values, representing the granularity and the size of the cells respectively, were measured by flow cytometry on the same cells and at the same time as the $C_{12}FDG$ fluorescence. Each bar shows the percentage of cells with the highest FSC and SSC values. Each condition was performed in triplicate. Each bar represents the mean ± SD of the three measures. *t*-tests were performed for each counting in comparison to nonirradiated exponentially growing NHDFs.

The online version of this article includes the following source data and figure supplement(s) for figure 2:

**Source data 1.** Data for SA-β-Gal activity.

**Source data 2.** Data for forward scatter (FSC) and side scatter (SSC levels).

**Figure supplement 1.** Normal human dermal fibroblasts (NHDFs) from another donor positioned at the margin of a planning target volume (PTV) receiving 2 Gy/day undergo senescence.

**Figure supplement 1—source data 1.** Values of $C_{12}FDG$ fluorescence intensity.

**Figure supplement 2.** Normal human dermal fibroblasts (NHDFs) positioned at the margin of the planning target volume (PTV) receiving 2 Gy/day become cell cycle arrested in G1.

**Figure supplement 2—source data 1.** Quantification of cells in the different phases of the cell cycle.

**Figure supplement 3.** Induction of p16 and p21 expression in normal human dermal fibroblasts (NHDFs) positioned at the margin of a planning target volume (PTV) receiving 2 Gy/day.

**Figure supplement 3—source data 1.** RT-qPCR data for p16.

**Figure supplement 3—source data 2.** RT-qPCR data for p21.

**Figure supplement 4.** Induction of p16 in normal human dermal fibroblasts (NHDFs) positioned at the margin of a planning target volume (PTV) receiving 2 Gy/day.

**Figure supplement 4—source data 1.** Western blot of p16.

**Figure supplement 4—source data 2.** Quantification of p16-positive cells.

**Figure supplement 5.** A significant proportion of normal human dermal fibroblasts (NHDFs) positioned at the margin of a planning target volume (PTV) receiving 2 Gy/day become multinucleated.

**Figure supplement 5—source data 1.** Quantification of multinucleated cells.

## NHDFs positioned at the margin of the PTV develop an SSB-without-DSB signature

Since the cell outcomes differed whether they were positioned inside or at the margin of the PTV, we next wanted to characterize potential quantitative or qualitative differences regarding the DNA breaks generated in the two areas. We firstly irradiated NHDFs cultured in 96-well plates once at 2 Gy and searched for SSBs and DSBs by performing an immunodetection of the signaling proteins XRCC1 for SSBs, and 53BP1 and γH2AX for DSBs. In NHDFs positioned inside the PTV, we detected both XRCC1 and 53BP1/γH2AX foci as expected. In NHDFs positioned at the margin, we detected almost only XRCC1 foci. Surprisingly, they were as much numerous in the margin as in the PTV, whatever the distance, up to 5 cm. In contrast, the quantity of 53BP1/γH2AX foci per cell rapidly dropped down with the distance from the limit of the PTV, as expected (***Figure 3A***, ***Figure 3—figure supplement 1***, ***Figure 3—figure supplement 2***).

To further challenge this surprising repartition of XRCC1 foci, we searched DNA breaks by comet assays, a single-cell electrophoresis technique that allows a direct detection of the physical breaks, in contrast to XRCC1 and 53BP1 immunofluorescences, which highlight DNA breaks once they are sensed and signalized by the cell. When the comet assay electrophoresis is performed at pH 12.3, the double helix is denatured and the DNA fragments that migrate and form the comet tail come from both SSBs and DSBs. When the electrophoresis is performed at pH 8, the DNA fragments that migrate come only from DSBs. Thus, comparing the results at pH 8 and 12.3 enables to evaluate the

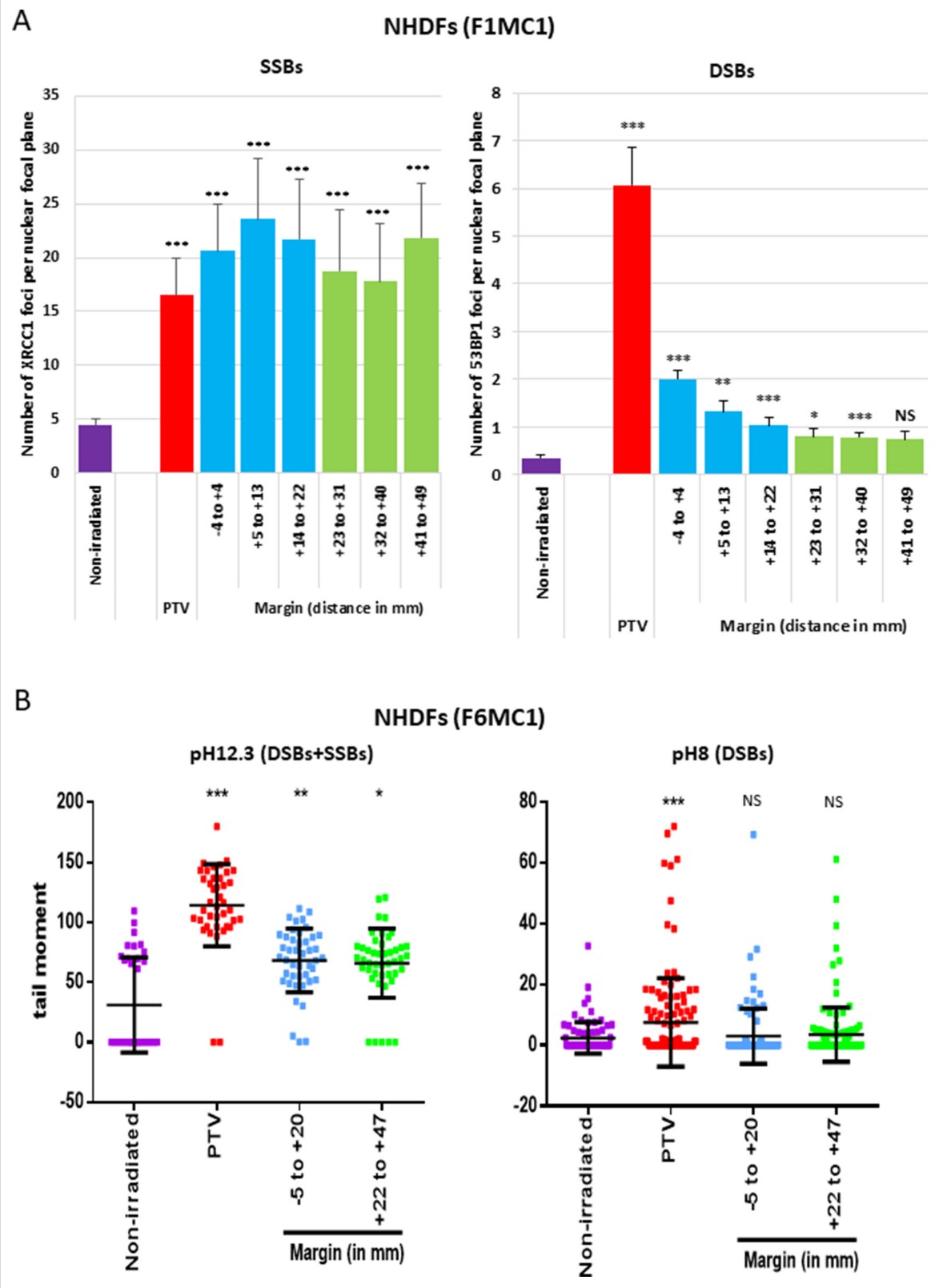

**Figure 3.** Single-strand breaks (SSBs) and double-strand breaks (DSBs) generated in normal human dermal fibroblasts (NHDFs) positioned at the margin or inside a planning target volume (PTV) having received 2 Gy. (**A**) Proliferating NHDFs (donor F1MC1, population doubling [PD] = 32) positioned straddling the PTV and the margin were irradiated once at 2 Gy. Cells were fixed 20 min or 2 hr later for the immunodetection of XRCC1 and 53BP1, respectively. XRCC1 and 53BP1 foci were quantified by high-content microscopy. The bar chart represents the mean number ± SD of foci per nuclear

*Figure 3 continued on next page*

*Figure 3 continued*

focal plane in 100 cells of all wells of a column. An ANOVA test with a Bonferroni correction was performed for each counting in comparison to nonirradiated exponentially growing NHDFs. (**B**) Proliferating NHDFs (F6MC1, PD = 20.53) were irradiated as in (**A**), immediately put at 10°C for 30 min and then processed for comet assays. Each dot in the scatter plots represents the value of the tail moment of one cell. The columns regroup the values of about 100 cells for each condition and give the mean ± SD of tail moment values. A Kruskal–Wallis test with a Bonferroni correction was performed for each counting in comparison to nonirradiated exponentially growing NHDFs.

The online version of this article includes the following source data and figure supplement(s) for figure 3:

**Source data 1.** Number of XRCC1 foci.

**Source data 2.** Number of 53BP1 foci.

**Source data 3.** Statistics on comet assays at pH 12.3.

**Source data 4.** Comet assays at pH 12.3.

**Source data 5.** Comet assays at pH 8.

**Source data 6.** Statistics on comet assays at pH 8.

**Figure supplement 1.** Representative images of XRCC1 and 53BP1 foci recorded by high-content microscopy.

**Figure supplement 2.** Normal human dermal fibroblasts (NHDFs) positioned at the margin of a planning target volume (PTV) receiving 2 Gy do not suffer from double-strand breaks (DSBs).

**Figure supplement 2—source data 1.** Quantification of γH2AX foci.

**Figure supplement 3.** Single-strand breaks (SSBs) and double-strand breaks (DSBs) generated in normal human dermal fibroblasts (NHDFs) derived from different donors positioned at the margin or inside a planning target volume (PTV) receiving 2 Gy.

**Figure supplement 3—source data 1.** Comet assays at pH 12.3 with F26MC1 NHDFs.

**Figure supplement 3—source data 2.** Statistics on comet assays at pH 12.3 with F26MC1 NHDFs.

**Figure supplement 3—source data 3.** Comet assays at pH 8 with F26MC1 NHDFs.

**Figure supplement 3—source data 4.** Statistics on comet assays at pH 8 with F26MC1 NHDFs.

**Figure supplement 3—source data 5.** Comet assays at pH 12.3 with F1MC1 NHDFs.

**Figure supplement 3—source data 6.** Statistics on comet assays at pH 12.3 with F1MC1 NHDFs.

**Figure supplement 3—source data 7.** Comet assays at pH 8 with F1MC1 NHDFs.

**Figure supplement 3—source data 8.** Statistics on comet assays at pH 8 with F1MC1 NHDFs.

**Figure supplement 4.** The numbers of 53BP1 and XRCC1 foci recorded after an irradiation with 6 and 20 MV photon beams are similar.

**Figure supplement 4—source data 1.** Number of XRCC1 and 53BP foci.

**Figure supplement 5.** A propagation of ionizing particles in the material is necessary for single-strand break (SSB) induction at distance from the planning target volume (PTV).

**Figure supplement 5—source data 1.** Number of XRCC1 foci.

**Figure supplement 5—source data 2.** Statistics on XRCC1 foci number.

**Figure supplement 5—source data 3.** Number of 53BP1 foci.

**Figure supplement 5—source data 4.** Statistics on 53BP1 foci number.

**Figure supplement 6.** The proliferating versus quiescent status of normal human dermal fibroblasts (NHDFs) does not interfere with the generation of single-strand breaks (SSBs).

**Figure supplement 6—source data 1.** Quantification of cells in the different phases of the cell cycle.

**Figure supplement 6—source data 2.** Comet assays.

relative quantity of DSBs and SSBs. Comet assay results at pH 12.3 revealed the presence of breaks in NHDFs positioned at the margin up to about 5 cm. Comet assay results at pH 8 confirmed the lack of DSB formation in NHDFs positioned at the margin (*Figure 3B*, *Figure 3—figure supplement 3*). Therefore, the breaks recorded in the margin correspond almost only to SSBs, hence confirming the immunofluorescence results. As in immunofluorescence results, SSBs were as numerous in the proximal and distal margins. Comet assay results also confirmed the formation of both SSBs and DSBs in NHDFs positioned inside the PTV (*Figure 3B*, *Figure 3—figure supplement 3*).

In order to target the tumor at best in its tridimensional volume, the planning treatment system combines the use of beams at 6 and 20 MV. To determine whether the energy of the incident photons could impact on the nature or quantity of breaks generated in NHDFs positioned inside the PTV or at its margin, we measured the quantity of XRCC1 and 53BP1 foci generated in NHDFs irradiated at 6

versus 20 MV. The results indicate that there was no statistically significant difference in the quantity of SSBs and DSBs generated inside the PTV or at the margin after one irradiation at 2 Gy at 6 or 20 MV (*Figure 3—figure supplement 4*).

Therefore, a unitary 2 Gy irradiation generates in nontumoral fibroblasts surrounding the PTV a specific SSB-without-DSB signature. As expected (*Hagen, 1994*), the quantity of the generated DSBs followed approximately the dose gradient that exponentially decreases from the PTV to the most distal margin. In contrast and surprisingly, the quantity of generated SSBs was almost constant from the PTV to the most distal margin assayed here (about 5 cm).

## The SSB generation in the margin necessitates some continuity in the material from the PTV

It was established that the major determinant of the dose deposited at the margin of a PTV results from the scattering of photons inside the patient (*Chofor et al., 2012*). To investigate whether the generation of SSBs in cells positioned in the margin was due to the propagation of photons inside the material, here the plastic of the culture plates, we aligned a second culture plate after the one straddling the limit of the PTV, and the two aligned plates were sealed or not by an ultrasound gel (*Figure 3—figure supplement 5A*). We irradiated once at 2 Gy and analyzed DNA breaks by immuno-fluorescence anti-XRCC1 and -53BP1. Regarding SSBs, we recorded XRCC1 foci in NHDFs positioned inside the PTV and at the margin in only the first plate when the two plates were not sealed, but also in the second plate, when the plates were sealed, up to 18.7 cm. Regarding DSBs, we recorded 53BP1 foci only in NHDFs positioned inside the PTV and, to a much less extent, in the most proximal margin of the first plate, confirming the previous results (*Figure 3—figure supplement 5B*). This result suggests that SSBs are generated by ionizing particles that propagate inside the material, here in the plastic of the culture plates.

## The SSB-without-DSB signature of the margin is not affected by the cycling versus quiescent status of NHDFs

All the above experiments were performed using proliferating NHDFs cultured in a basal medium complemented by 2% fetal bovine serum (FBS), FGF, and insulin. However, in vivo in the dermis, fibroblasts are rather quiescent, except during tissue repair. We therefore wanted to determine whether the quantity of DNA breaks recorded after irradiation could be affected by the proliferating versus quiescent status of NHDFs. Cell cycle-slowed-down NHDFs (cultured in 0.1% FBS) and proliferating NHDFs (cultured in 2% FBS) were irradiated at 2 Gy daily for 5 days. The amount of DSBs plus SSBs was quantified by comet assays at pH 12.3. The results indicate that there was no difference in SSB plus DSB quantity generated in cells positioned in the margin when cultured in 0.1% FBS compared to 2% (*Figure 3—figure supplement 6*).

## The SSB-without-DSB signature of the margin is also recorded in vivo

To go further on the physiopathological relevance of the SSB-without-DSB signature of the margin, we quantified the marginal DNA damages in mice. For that, mice were positioned in close contact (sealed with an ultrasound gel) to a phantom that was irradiated using the Elekta Precise of the OncoVet veterinary clinic (Villeneuve d'Ascq, France). This linear accelerator is very similar to the Varian CLINAC of the Anti-Cancer Center of Lille we used for irradiating cells. The isocenter was positioned at the level of the mouse skin. A square PTV of 25 × 25 cm was delimited. The phantom was positioned inside the PTV, aligned along one of its limits, and the mouse was glued to it using some ultrasound gel (*Figure 4A*). The phantom was irradiated once at 2 Gy. Mice were sacrificed 1 hr later, their skin collected, fragmented at different distances from the contact with the phantom, and the skin fragments processed for detecting XRCC1 and 53BP1 foci. Mice irradiated whole body (isocenter at the level of the skin) were used as positive control. As expected, both SSBs and DSBs developed in dermal cells of total body-irradiated mice. In contrast, only SSBs, but not DSBs, were generated in dermal cells of the skin of mice positioned at the margin of the phantom, up to 2 cm from the contact with the phantom (*Figure 4B*). By co-detecting vimentin with XRCC1, we showed that most of the XRCC1-positive dermal cells were fibroblasts (*Figure 4—figure supplement 1*). Therefore, the peri-PTV in vivo tissues also specifically develop an SSB-without-DSB signature upon irradiation.

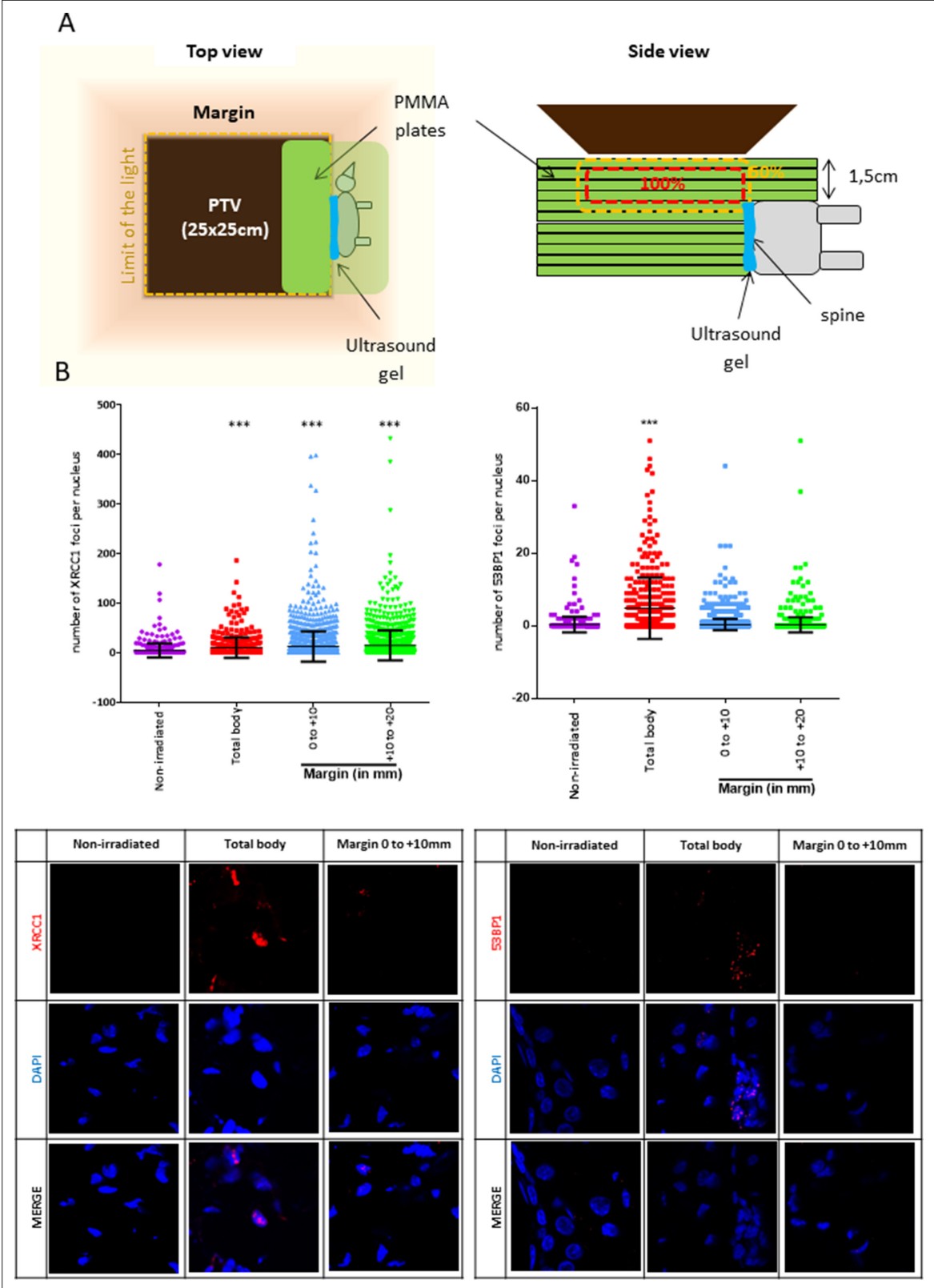

**Figure 4.** Mice positioned at the margin of a phantom mimicking a planning target volume (PTV) irradiated once at 2 Gy have a significant increase of XRCC1 foci. (**A**) Schematic representation of how mice were positioned at the margin of a phantom. (**B**) Six mice (three females and three males) were irradiated as in (**A**), one mouse was irradiated positioned inside the PTV (positive control designed as 'total body') and four mice (two females and two males) were not irradiated but manipulated as the others. Mice were sacrificed 1 hr after irradiation. Skin samples at different distances from

*Figure 4 continued on next page*

*Figure 4 continued*

the isodose 50 (the spine), established as the zero, were dissected and fixed for the immunodetection of XRCC1 and 53BP1. XRCC1 and 53BP1 foci of dermal cells were quantified by confocal microscopy. One skin sample per distance and per mice was used. Inside each sample, about 200 cells were analyzed. Each dot in the scatter plots represents the number of XRCC1 or 53BP1 foci of one dermal cell. The columns regroup all the values for each condition and give the mean ± SD of foci number values. A Kruskal–Wallis test with a Bonferroni correction was performed for each counting in comparison to nonirradiated mice. The lower panel shows representative fluorescent microscopic images of XRCC1 or 53BP1 foci in red and nucleus stained by DAPI in blue.

The online version of this article includes the following source data and figure supplement(s) for figure 4:

**Source data 1.** Number of XRCC1 foci.

**Source data 2.** Statistics on XRCC1 foci numbers.

**Source data 3.** Number of 53BP1 foci.

**Source data 4.** Statistics on 53BP1 foci numbers.

**Figure supplement 1.** Mice positioned at the margin of a phantom mimicking a planning target volume (PTV) irradiated once at 2 Gy have a significant increase of XRCC1 foci in dermal fibroblasts.

**Figure supplement 1—source data 1.** Quantification of XRCC1 foci.

**Figure supplement 1—source data 2.** Statistics on the quantification of XRCC1 foci.

## SSBs generated in NHDFs positioned at the margin accumulate with successive radiation fractions because of an exhaustion of the repair capacity

A standard conformal 3D radiation therapy regimen in France is made of radiation fractions of 2 Gy delivered every day, except weekends. We wondered whether the DNA breaks generated in the PTV and at the margin could accumulate along the successive radiation fractions of such a therapeutic protocol. To address this question, we first compared by comet assays the quantities of SSBs and DSBs after only one irradiation to that after the last irradiation of a series of five. Regarding DSBs in NHDFs positioned inside the PTV, the results of comet assays at pH 8 show that there was no significant difference between cells irradiated once or five times. Regarding DSBs in NHDFs positioned at the margin, the results confirmed again their almost absence. Regarding SSBs, the comparison of the results at pH 8 and 12.3 indicated that in NHDFs positioned inside the PTV, SSBs were less numerous after five irradiations than after just one. In contrast, in NHDFs positioned at the margin, SSBs were more numerous after five irradiations than after just one (*Figure 5*). Therefore, the SSBs generated in NHDFs positioned at the margin accumulate with successive radiation fractions, in contrast to both DSBs and SSBs generated in cells positioned inside the PTV that do not.

The above results could be explained by a decrease in the SSBR capacity with successive radiation fractions. To assay this hypothesis, we submitted NHDFs to a unique irradiation at 2 Gy or to a series of five, and we performed the comet assays either just after the last irradiation or after a 72 hr delay, to evaluate whether the repair has happened or not. The results indicate that SSBs formed in cells positioned either in the PTV or the margin were fully repaired within 72 hr after a unique irradiation, whereas they were still detected 72 hr after the last irradiation of a series of five (*Figure 5—figure supplement 1*), indicating that the SSB repair capacity decreased with successive radiation fractions. In contrast, the DSBs were always fully repaired within 72 hr, whatever the cells were submitted to one or five radiation fractions (*Figure 5—figure supplement 1*), indicating that the DSB repair capacity was not affected along a fractionated irradiation protocol. The same results were obtained by analyzing the breaks by immunofluorescence against XRCC1 and 53BP1: XRCC1 foci (SSBs) generated in the PTV were also partly resolved within 72 hr, but those generated in the margin were still significantly present after 72 hr; 53BP1 foci (DSBs) present in NHDFs positioned inside the PTV or at the proximal margin after four successive radiation fractions at 2 Gy were partly resolved within 72 hr (*Figure 5—figure supplement 2*). These results confirm that the SSBR capacity declines with the successive radiation fractions of a therapeutic protocol.

Since this decline could be due to an exhaustion of the pool of NAD+ (the substrate of PARP1) because of the high and continuous demand, we evaluated the evolution of the PARylation capacity of the cells along a fractionated protocol. NHDFs were irradiated at 2 Gy, once or during 2 weeks, and just after the last radiation fraction they were challenged by $H_2O_2$ to evaluate their remaining capacity

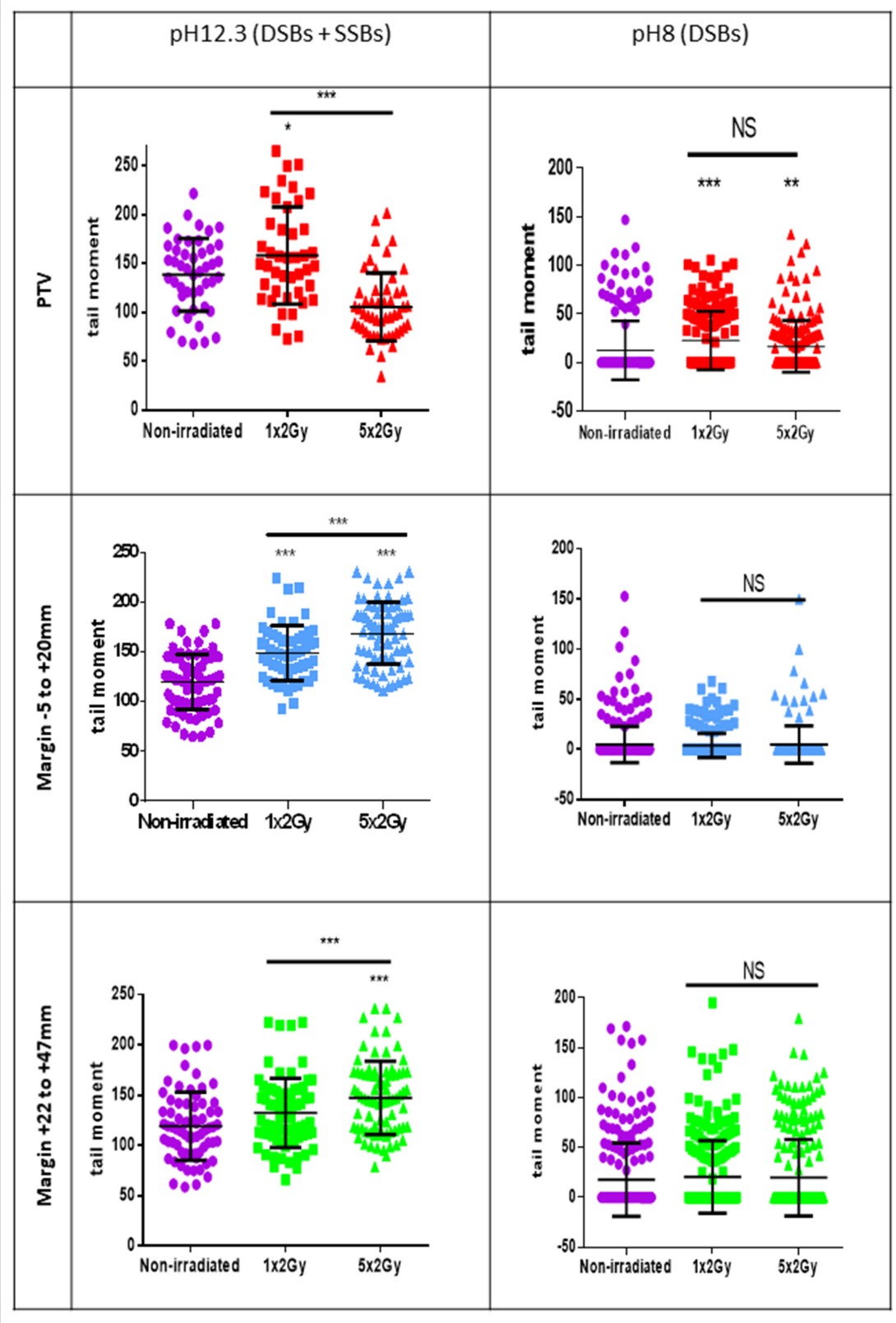

**Figure 5.** Single-strand breaks (SSBs) accumulate with successive irradiations in normal human dermal fibroblasts (NHDFs) positioned at the margin of a planning target volume (PTV) receiving 2 Gy/day. Proliferating NHDFs (donor F6MC1, population doubling [PD] = 22.67) cultured in 12-well plates were irradiated or not only one time at 2 Gy or during 5 days at 2 Gy daily. Just after the last radiation fraction, cells were put at 10°C for 30 min and then processed for comet assays. About 150 cells per condition were analyzed. Each dot in the scatter plots represents the value of the tail moments of

*Figure 5 continued on next page*

*Figure 5 continued*

one cell. The columns regroup all the values for each condition and give the mean ± SD of tail moment values. A Kruskal–Wallis test with a Bonferroni correction was performed for each counting in comparison to nonirradiated exponentially growing NHDFs or between cells irradiated one or five times.

The online version of this article includes the following source data and figure supplement(s) for figure 5:

**Source data 1.** Comet assays at pH 8 for the planning target volume (PTV).

**Source data 2.** Statistics on comet assays at pH 8.

**Source data 3.** Statistics on comet assays at pH 12.3.

**Source data 4.** Comet assays at pH 8 for the margin –5 + 20.

**Source data 5.** Comet assays at pH 8 for the margin +22 + 47.

**Source data 6.** Comet assays at pH 12.3 for the planning target volume (PTV).

**Source data 7.** Comet assays at pH 12.3 for the margin –5 + 20.

**Source data 8.** Comet assays for the margin +22 + 47.

**Figure supplement 1.** Single-strand breaks (SSBs) generated in normal human dermal fibroblasts (NHDFs) positioned at the margin of the planning target volume (PTV) are no more repaired after five successive irradiations at 2 Gy.

**Figure supplement 1—source data 1.** Comet assays pH 12.3.

**Figure supplement 1—source data 2.** Statistics on comet assays pH 12.3.

**Figure supplement 1—source data 3.** Comet assays pH 8.

**Figure supplement 1—source data 4.** Statitstics on comet assays pH 8.

**Figure supplement 2.** The repair of single-strand breaks (SSBs) generated in normal human dermal fibroblasts (NHDFs) positioned at the margin of the planning target volume (PTV) becomes poorly efficient after four successive irradiations at 2 Gy.

**Figure supplement 2—source data 1.** Quantification of XRCC1 foci.

**Figure supplement 2—source data 2.** Quantification of 53BP1 foci.

**Figure supplement 3.** The PARylation capacity decreases with successive irradiations.

**Figure supplement 3—source data 1.** Quantification of PAR foci.

**Figure supplement 3—source data 2.** Statistics on quantification of PAR foci.

in synthesizing PAR chains. Immunofluorescence against PAR chains shows that after one irradiation NHDFs do synthesize PAR chains in response to $H_2O_2$, whereas after 10 successive radiation fractions they have become unable to do so (*Figure 5—figure supplement 3*).

Taken together, these results suggest that the successive irradiations of a standard fractioned radiotherapy protocol could lead to an exhaustion of the PARylation capacity, provoking the accumulation of unrepaired SSBs.

## A few radio-induced senescent cells generated in the margin give rise to transformed daughter cells

As unrepaired SSBs could be mutagenic, we wondered whether the marginal senescent cells could be a source of (pre)-neoplastic cells. To investigate this point, we irradiated NDHFs positioned inside the PTV or at its margin at 2 Gy/day during 2 weeks up to the induction of senescence. Since all cells located in the margin were probably not fully senescent at the end of this treatment, we sorted the most senescent NHDFs according to their SA-β-Gal activity, size, and granularity (*Figure 6A*, *Figure 6— figure supplement 1*). Sorted cells were plated again and monitored for post-senescence emergence. After 2–4 weeks post-sorting according to the experiment, we noticed the reappearance of small proliferating cells amongst senescent cells emanating from the margin (*Figure 6A and B*, *Figure 6— figure supplement 2A*) referred to as post-senescent neoplastic emergent (PSNE). To determine whether these cells displayed (pre-)neoplastic properties, we first assayed whether they were mutated by performing an Hypoxanthine Phosphorybosyl Transferase (HPRT assay). As expected, exponentially growing cells did not survive a 6-thioguanine (6-TG) treatment, meaning their *hprt* gene, taken as a reporter of the genome condition, was not loss-of-function mutated. In contrast, PSNE cells, having emerged from senescent cells of the margin, resisted the 6-TG treatment, suggesting they were carrying mutations (*Figure 6C*, *Figure 6—figure supplement 2B*). Then, we evaluated the invasive properties of the PSNE cells. We first searched for a potential matrix metalloproteinase (MMP)

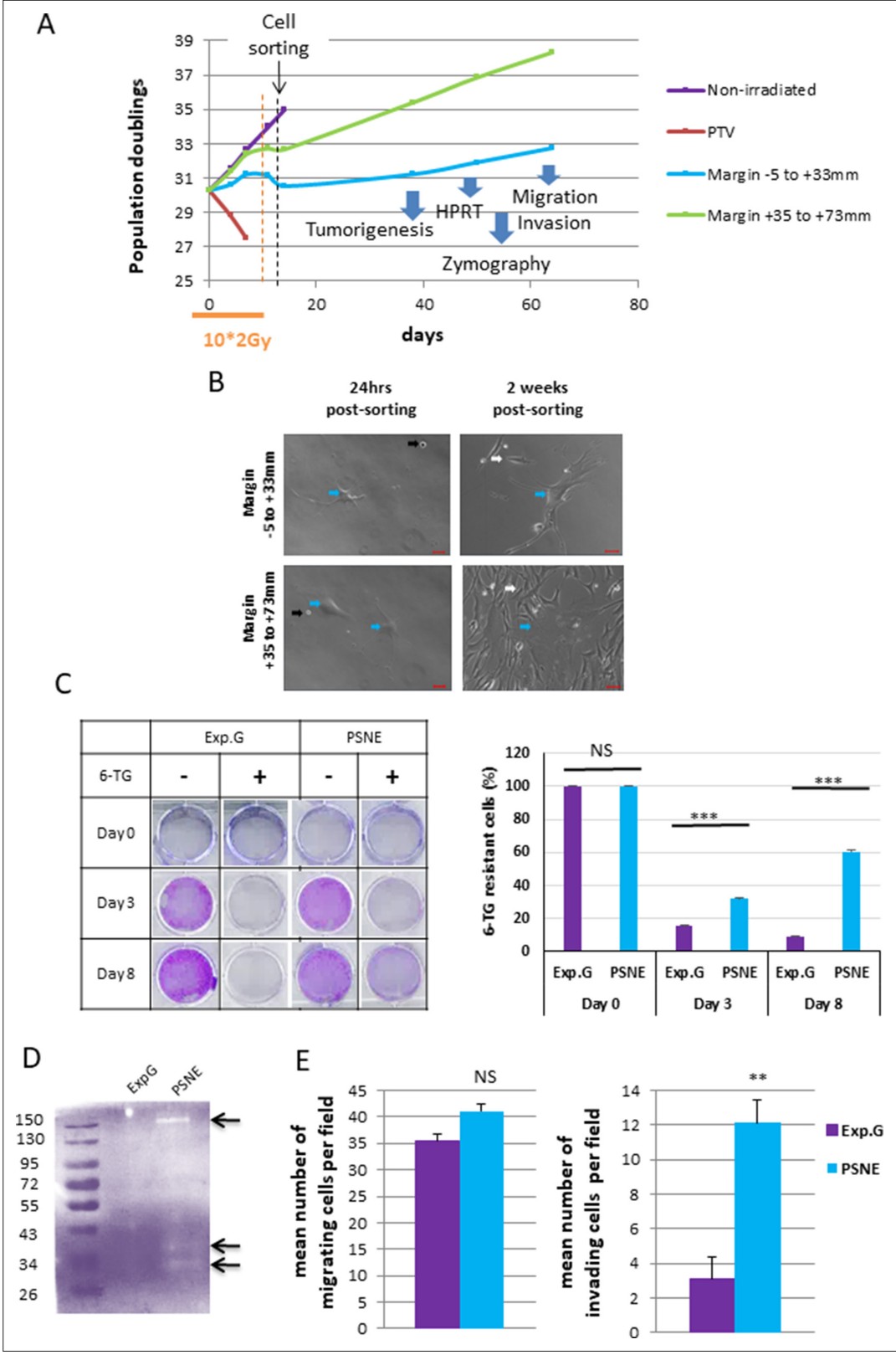

**Figure 6.** Some radio-induced senescent cells generated in the margin escape from senescence to generate a progeny of cells that reproliferate, have an increased invasion capacity, and display mutations. (**A**) Normal human dermal fibroblasts (NHDFs) (donor F6MC1, population doubling [PD] = 29.6 at the beginning of the experiment) positioned straddling the planning target volume (PTV) and the margin were irradiated or not during 2 weeks (10

*Figure 6 continued on next page*

*Figure 6 continued*

\* 2 Gy). Cells were counted at each passage using a Malassez chamber. After 2 weeks of irradiation, the senescent cells generated in the margin were sorted according to their size, granularity, and SA-β-Gal activity (see parameters of sorting in *Figure 6—figure supplement 1*). After sorting, cells were put again in culture and followed at long term. (**B**) Phase-contrast microscopic images of cells of the margin 24 hr and 2 weeks post-sorting. The blue arrows indicate cells with a senescent morphology. The black arrows indicate cells that have not replated. The white arrows indicate small reproliferating cells. (**C**) Hypoxanthine Phosphorybosyl Transferase (HPRT) assays performed on cells having regrown from sorted senescent cells generated in the margin referred to as post-senescence neoplastic emerging (PSNE) cells and on nonirradiated exponentially growing NHDFs at 17 PDs (used as control for normal cells). The bars indicate the mean ± SD of three measures of the percentage of 6-thioguanine (6-TG)-resistant cells. (**D**) Metalloproteinase (MMP) activity analysis of culture supernatant from PSNE or exponentially growing cells by 0.1% gelatin zymography. Arrows show bands of digested gelatin. (**E**) Migrating and invasive capacities were assayed in Boyden chambers without or with Matrigel, respectively. Cells that have passed across the filters were counted in five fields per chamber in three different chambers for each condition. Each bar represents the mean ± SD of the number of cells per field. *t*-tests were performed for comparing exponentially growing and PSNE cells.

The online version of this article includes the following source data and figure supplement(s) for figure 6:

**Source data 1.** Data of growth curves.

**Source data 2.** Quantification of 6-thioguanine (6-TG)-resistant cells.

**Source data 3.** Zymography.

**Source data 4.** Quantification of cell migration and invasion.

**Figure supplement 1.** Principle of senescent cell sorting.

**Figure supplement 2.** Similar experiment as in *Figure 6* with normal human dermal fibroblasts (NHDFs) derived from another donor.

**Figure supplement 2—source data 1.** Growth curves.

**Figure supplement 2—source data 2.** Quantification of 6-thioguanine (6-TG)-resistant cells.

activity by in-gel zymography. Gelatinase activity was detected in PSNE cells emanating from NHDFs positioned in the margin compared to exponentially growing NHDFs taken as control (*Figure 6D*). Then, we assayed their migrating and invading capacities in Boyden chamber assays using respectively Matrigel-uncoated or -coated filters. PSNE cells, having emerged from senescence induced in the margin, did not show any increased migration compared to nonirradiated exponentially growing cells. However, their invasive capacities were multiplied about fourfold compared to control cells (*Figure 6E*). Finally, we evaluated their tumorigenic potential by performing xenografts in immunocompromised mice (CB17-SCID). Twenty-two mice were grafted with $10^6$ PSNE cells emanating from NHDFs positioned in the margin (proximal or distal). Up to now, 10 months after grafting, none of these mice developed tumors, whereas all mice grafted with HT1080 cells, a sarcoma cell line, developed a tumor within 2 weeks. Taken together, these results evidence that along irradiations of a fractionated protocol normal fibroblasts of the vicinity of a PTV enter premature senescence from which, after a while, few rare cells evade to give rise to a progeny of proliferative, mutated, invading, but not fully tumorigenic cells.

## An SSB-only-accumulation is sufficient to induce both senescence and neoplastic escape

Ionizing radiations induce SSBs and DSBs, but also several other damages to DNA or other cell components (*Reisz et al., 2014*). Therefore, we wondered if the sole SSB accumulation in the absence of DSBs or other DNA damages could be sufficient to induce senescence and post-senescence neoplastic escape. To address this question, we set a protocol to specifically and solely induce unrepaired SSBs, independently of an irradiation. For that, we daily treated NHDFs with a very mild 5 μM concentration of hydrogen peroxide ($H_2O_2$) to induce SSBs. To make them accumulate, we inhibited their repair by co-treating the cells with ABT-888, also called veliparib, a specific inhibitor of PARP1 and PARP2 (*Donawho et al., 2007*). We checked that the combined treatment did induce a significant accumulation of XRCC1 foci, whereas the treatments with $H_2O_2$ alone or veliparib alone were very much less efficient. The accumulation of XRCC1 foci began from the 14th day of the combined treatment

(*Figure 7A*). We also checked that the combined treatment did not induce either an accumulation of DSBs, oxidative base damages visualized by OGG1 foci, or oxidative damage to mitochondria analyzed with the JC1 probe (*Figure 7—figure supplement 1*). We also analyzed the phosphorylation of XRCC1 and the recruitment of PNKP at the damage foci to check the SSBR blockage by their absence (*Figure 7—figure supplement 2*). We then followed up the consequence of the accumulation of unrepaired SSBs on the cell behavior. For that, we stopped the $H_2O_2$ treatment at day 21 of the combined treatment, but maintained at long term the veliparib treatment, to maintain the SSBs unrepaired. We observed a rise in the number of SA-β-Gal-positive cells (*Figure 7B*, *Figure 7—figure supplement 3A, B*, *Figure 7—figure supplement 4B*). This SA-β-Gal activity rising was correlated with a cell growth arrest from the 25th day (*Figure 7C*, *Figure 7—figure supplement 3A*), an increased cell size, spreading and granularity (*Figure 7D*, *Figure 7—figure supplement 4A*), an upregulation of p16 expression, and a hypophosphorylation of Rb (*Figure 7—figure supplement 4C*). These data attest that a specific accumulation of SSBs only results in premature senescence. Notice that the $H_2O_2$ treatment also induced an increase in cell death (*Figure 7—figure supplement 4D*).

We then monitored the outcome of the SSB-induced senescent cells. We observed a growth recovery from the days 35–50 according to the experiment (*Figure 7C*, *Figure 7—figure supplement 3A*) due to the appearance in the culture dishes of small cells with a stellate shape (*Figure 7D*), which were SA-β-Gal negative (*Figure 7B*) and resistant to 6-TG in HPRT assays (*Figure 7E*).

All these results suggest that the sole SSB accumulation in the absence of DSBs or other damages is sufficient to induce a premature senescent state from which a few cells escape to generate reproliferating, morphologically transformed and mutated daughter cells.

## Discussion

Second sarcomas are a rare but very severe late side effect of radiation therapy. Several studies conclude that they could be due to the out-of-field dose affecting normal cells surrounding the treated volume (*Chofor et al., 2012*; *Chofor et al., 2011*). In this study, we investigated the effects of this out-of-field dose on normal fibroblasts in terms of DNA damage and cell outcome. We show that the out-of-field dose induces the formation of SSBs, but nearly no DSBs. Importantly, the SSBs accumulate during a mimicked standard therapeutic protocol because of a lack of repair due to a decrease in the PARylation capacity that establishes along the successive daily radiation fractions. The fibroblasts affected by this specific accumulation of unrepaired SSBs do not undergo cell death but enter in premature senescence. They remain stably arrested in this senescent state for a while, and then a few of them re-enter cell cycle to generate a progeny of daughter cells displaying hallmarks of cancerous transformation, but which have not become fully tumorigenic. This scenario was established by robust in vitro experiments using proliferating or quiescent primary human fibroblasts derived from different donors, as well as by in vivo experiments in mice. It is, to our knowledge, the first scenario that could explain at the same time the latency period of second sarcoma emergence, underpinned by the long life of senescent cells, and the preferential location of second sarcomas around the treated volume, underpinned by the specific accumulation in this area of nonlethal but mutagenic unrepaired DNA damages.

It is known that the marginal dose exponentially decreases with the distance from the limit of the beam (*Chofor et al., 2012*; *Kry et al., 2007*; *Wiezorek et al., 2009*). In accordance, we found that the quantity of DSBs dramatically decreases from a few millimeters from the limit of the beam. In contrast and surprisingly, SSBs were almost as numerous at several centimeters from the limit of the beam as inside the PTV. Moreover, we demonstrated that their formation necessitates continuity in the material from the PTV. This indicates that SSBs are induced by photons propagating inside the material, confirming several studies showing that the internal scattering is the major source of the dose deposited in the most proximal margin (*Chofor et al., 2012*; *Lee et al., 2016*). Moreover, it can be hypothesized that the preferential generation of SSBs on DSBs in the margin could be the consequence of the tightening of the scattered photon spectral energy fluence distribution around lower energies (*Chofor et al., 2012*; *Chofor et al., 2011*; *Kirkby et al., 2007*). The experimental setup of this study was designed to only address the question of the potential role of photons scattering from the PTV, without mixing with the bystander effect relying on molecules secreted by the irradiated cancer and normal cells present inside the PTV. Of course, in the patient context, these two mechanisms could

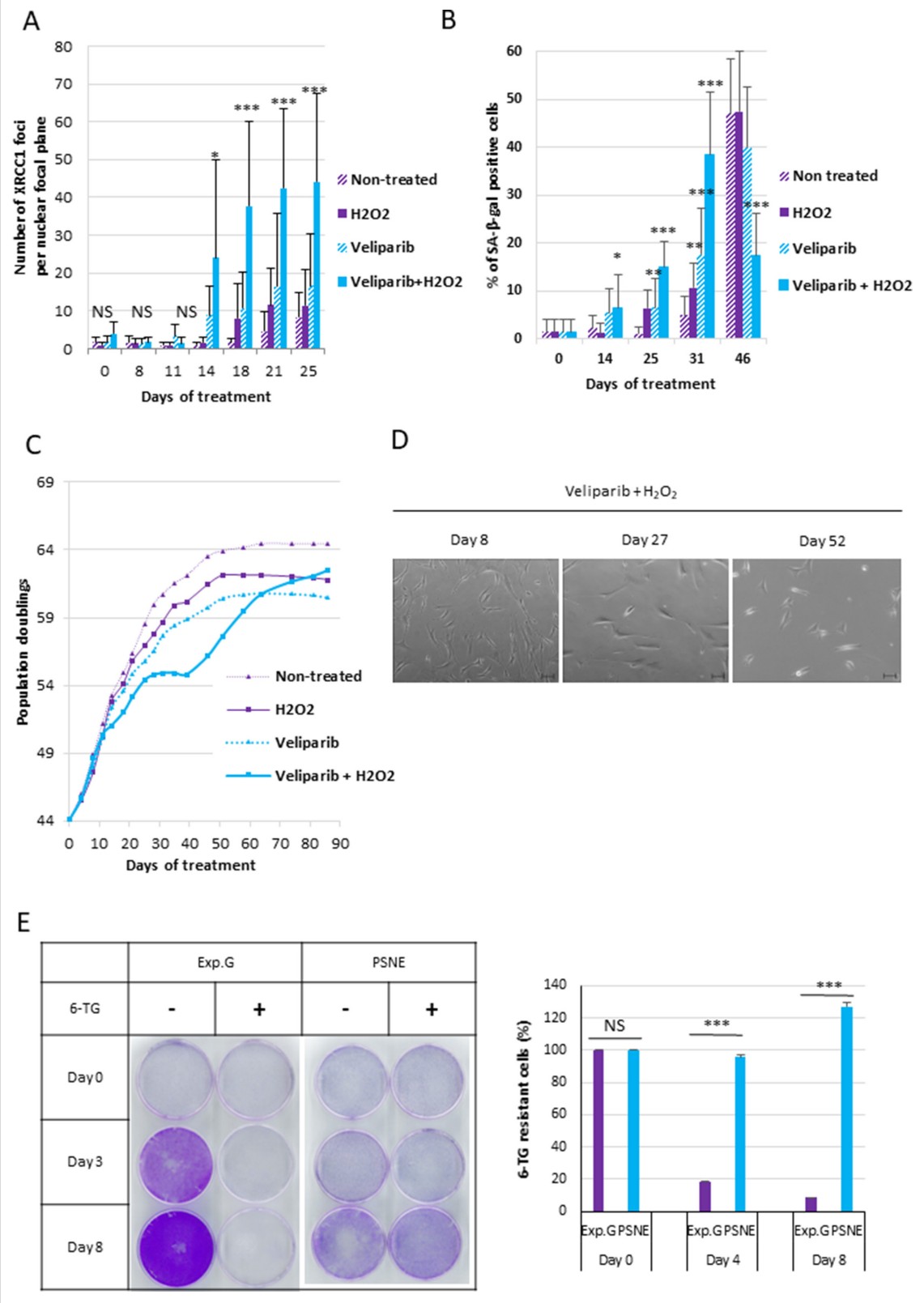

**Figure 7.** A single-strand break (SSB) accumulation is enough to induce senescence and post-senescence neoplastic escape (PSNE). (**A**) Analysis of XRCC1 foci by immunofluorescence in normal human dermal fibroblasts (NHDFs) (F1MC1, population doubling [PD] = 44.07 at the beginning of the experiment) treated daily or not with veliparib at 1 µM in combination with $H_2O_2$ at 5 µM during 21 days and then with veliparib alone. Each point represents the mean number of XRCC1 foci per nuclear focal plane at indicated time of treatment from 100 cells per condition. *t*-tests were performed

*Figure 7 continued on next page*

*Figure 7 continued*

for comparing nontreated and treated cells for each time point. (**B**) SA-β-Gal assays. Each bar represents the mean ± SD percent of SA-β-Gal-positive cells in an X-gal assay from five independent microscopic fields. *t*-tests were performed for comparing nontreated and treated cells for each time point. (**C**) Growth curves of the same cells. (**D**) Representative phase-contrast microscopy images of cell morphologies. (**E**) Hypoxanthine Phosphorybosyl Transferase (HPRT) assays performed on PSNE cells having regrown after the senescent plateau and on nontreated exponentially growing NHDFs (Exp. Growth) at 22 PDs as control. The bars indicate the mean ± SD of three measures of the percentage of 6-thioguanine (6-TG)-resistant cells. *t*-tests were performed for comparing exponentially growing cells with emerging cells.

The online version of this article includes the following source data and figure supplement(s) for figure 7:

**Source data 1.** Number of XRCC1 foci.

**Source data 2.** Quantification of SA-β-Gal assays.

**Source data 3.** Growth curves.

**Source data 4.** Quantification of 6-thioguanine (6-TG)-resistant cells.

**Figure supplement 1.** The combined poly(ADP)ribose polymerase (PARP) inhibitor + $H_2O_2$ treatment does not induce other oxidative damages than single-strand breaks (SSBs).

**Figure supplement 1—source data 1.** Western-blots.

**Figure supplement 1—source data 2.** Quantification of 53BP1 foci.

**Figure supplement 1—source data 3.** Quantification of hOGG1 foci.

**Figure supplement 1—source data 4.** Quantification of red on green fluorescence.

**Figure supplement 2.** The repair of single-strand breaks (SSBs) is blocked during a veliparib + $H_2O_2$ treatment.

**Figure supplement 2—source data 1.** Quantification of phosphoXRCC1 foci.

**Figure supplement 2—source data 2.** Quantification of polynucleotide kinase phosphatase (PNKP) foci.

**Figure supplement 3.** Same experiment as in *Figure 7* with another normal human dermal fibroblast (NHDF) donor.

**Figure supplement 3—source data 1.** Growth curves.

**Figure supplement 3—source data 2.** Quantification of SA-β-Gal assays.

**Figure supplement 4.** The combined poly(ADP)ribose polymerase (PARP) inhibitor + $H_2O_2$ treatment induces a panel of senescence markers.

**Figure supplement 4—source data 1.** Values of $C_{12}FDG$ fluorescence intensity.

**Figure supplement 4—source data 2.** Western blots.

**Figure supplement 4—source data 3.** Results of Annexin V/propidium iodide assays.

---

add, even synergize, or perhaps oppose the induction of DNA damages and the full transformation of the cells.

The results of this study also suggest that the ratio of DSBs/SSBs is very important in the cell outcome determination. Indeed, our data show that cells, whether they are cancerous or normal, developed both DSBs and SSBs when positioned inside the PTV, as already known (*Reisz et al., 2014*), and consequently either underwent apoptosis in a few days, or more slowly entered a senescent-like state and then died. In contrast, normal cells positioned at the margin, which developed almost exclusively SSBs, or normal cells in which we induced almost exclusively SSBs by a co-treatment with a mild concentration of $H_2O_2$ associated with a PARP inhibitor did not die at a significant level but entered senescence. Similarly, in normal human epithelial cells (keratinocytes and mammary epithelial cells), senescence occurs following an increase in endogenous oxidative stress, which also results in an accumulation of unrepaired SSBs, in the absence of DSBs or telomere shortening (*Nassour et al., 2016*). Very importantly, in all these situations where senescence is induced by SSBs in the almost absence of DSBs, telomere shortening or any other DNA damage able to induce the DDR pathway, a few cells systematically re-enter in S-phase to generate a progeny of daughter cells displaying neoplastic or pre-neoplastic properties (*Deruy et al., 2010*; *Gosselin et al., 2009b*; *Martin et al., 2014*; *Nassour et al., 2016* and results herein). In contrast, NHDFs (the same as those used in the irradiation experiments of this study), having entered in replicative senescence because of shortened telomeres, never undergo apoptosis and remain very stably arrested in G1 without ever escaping (*d'Adda di Fagagna et al., 2003*; *Nassour et al., 2016*). This state of replicative senescence fits very well with the dogma equating senescence with tumor suppression (*Campisi, 2005*; *Campisi, 2001*; *Kirkland and Tchkonia, 2017*; *Shay and Roninson, 2004*). In contrast, SSB-only-induced senescence,

which harbors a cell cycle arrest that is not so stable for a few cells and source of cells harboring cancerous hallmarks, has to be considered as a tumor-promoting state.

Senescence establishment implies the accumulation of DNA breaks, some of which have to remain unrepaired to sustain a prolonged activation of the cell cycle arrest pathways. This is true for replicative senescence in which shortened telomeres, which are assimilated to unrepaired DSBs, constantly activate the DDR pathway and downstream the tumor suppressor TP53 and its target p21, a CKI (*Rossiello et al., 2014*). This is also true for SSB-only-induced senescence of human keratinocytes and mammary epithelial cells where unrepaired SSBs activate the upregulation of p16 (*Nassour et al., 2016*), a major in vivo and in vitro senescence-associated CKI (*He and Sharpless, 2017*; *Rayess et al., 2012*). We now demonstrate this mechanism in this study also for fibroblasts in which SSBs were induced by a mild $H_2O_2$ treatment associated with a PARP inhibitor. And most importantly, we now demonstrate this mechanism for cells submitted to out-of-field radiations in a radiotherapy-mimicking context. But why SSBs are not repaired in this radiotherapy context is not yet completely established. We established in this study that the SSB repair capacity declines with the daily irradiations of a fractioned protocol, in correlation with a decline in the PARylation capacity. However, the precise mechanism of the PARylation capacity decline is presently unknown and is under investigation.

One major characteristic of cancer cells is their multiple mutations affecting oncogenes and tumor suppressor genes. The (pre-)neoplastic cells that are formed by SSB-only-induced senescence evasion display mutations and transformed characteristics including morphological change and increased invasion capacity (*Nassour et al., 2016* and results herein). The accumulation of unrepaired SSBs is the inducer of senescence, but paradoxically it is also the fuel for post-senescence evasion (*Nassour et al., 2016*), probably because of the mutagenic potential of unrepaired DNA breaks. However, the precise mechanisms by which SSBs induce senescence and (pre)neoplastic evasion remain to be investigated.

The results of this study open at least two therapeutic avenues to decrease the risk of developing a second cancer after radiotherapy. One could be to decrease the accumulation of unrepaired SSBs in peri-PTV cells by supplementing the patient with NAD+ precursors. However, this also should decrease the accumulation of SSBs in cancer cells, consequently decreasing their rate of death and thereby compromising the cure. The other avenue could be to try to eliminate the senescent cells after the end of the radiotherapy. It is now widely established that senescence contributes to multiple age-related dysfunctions and diseases, and eliminating senescent cells in mice delays or decreases the incidence of these disorders, including cancer (*Baker et al., 2011*; *Wissler Gerdes et al., 2020*). This prompted several teams to develop drugs, called senolytics, specifically targeting senescent cells, to alleviate age-related disorders. There are presently two major senolytics whose efficacy was demonstrated in vitro and in mouse models, and which are subject to clinical trials (*Chang et al., 2016*; *Pan et al., 2017*): (i) the BH3 mimetics ABT-737 and ABT-263 (navitoclax) and (ii) the association of dasatinib + quercetin. Interestingly, navitoclax, dasatinib, and quercetin are or were used in anticancer treatments, presupposing that their use after radiation therapy to prevent from a second cancer emergence should not hamper the efficacy of the radiotherapy, even they could act additionally or synergistically with the treatment.

To conclude, in addition to have a clinical interest, studying post-radiotherapy second sarcoma is a model to study the very initial steps of tumorigenesis. Indeed, most often, these steps are inaccessible because of affecting too few cells, of unknown type, altered by an unknown stressing or damaging agent. In the context of post-radiotherapy second sarcomas, the affected cells and the inducer are known. The very first tumorigenic event we highlight in this study involves the formation of discreet DNA damages – SSBs – often assumed to be of no consequence because these are easily repaired and do not affect the genome stability. We show here that cells adapt to these damages and survive for a long time by adopting the senescent phenotype. They thereby become a reservoir of potentially tumorigenic mutations.

## Materials and methods
### Cell cultures

NHDFs were purchased from PromoCell. Batches from three different donors were used in this study: F1MC1 from a male Caucasian donor 1 year old (lot number 6062703.3), F6MC1 from a male Caucasian

donor 6 years old (lot number 0080303.2), and F26FC1 from a female Caucasian donor 26 years old (lot number 415Z005.3). The cells were phenotypically characterized by the supplier by immuno-histochemical staining with a fibroblast-specific surface antigen. The supplier also assayed them for microbial contaminants (bacteria/fungi and mycoplasma) and infectious viruses (HIV-1, HBV, HCV). Cells were grown at 37°C in an atmosphere of 5% $CO_2$ and at the atmospheric $O_2$ tension. The culture medium was the FGM-2 bulletkit medium (CC-3132, Lonza). It consists in Fibroblast Basal Medium (FBM, CC-3131 from Lonza) supplemented by 2% FBS, human fibroblast growth factor, insulin at 5 mg/mL, gentamicin at 50 µg/mL, and amphotericin at 50 µg/mL. Cells were seeded at 3500 cells/$cm^2$ and subcultured at 70% confluence. The number of population doublings (PDs) was calculated at each passage by using the following equation: PD = log (number of collected cells/number of plated cells)/log2.

MDA-MB-231 were purchased from the European Collection of Authenticated Cell Cultures (92020424, ECACC). They were confirmed by the supplier as human Caucasian breast adenocarcinoma cells and given as devoid of infectious virus or toxic products. They were cultivated in DMEM (high glucose: 4.5 g/L) medium (41965039, Lifetech) supplemented by 1 mM sodium pyruvate (11360070, Gibco), 10% FBS (CVFSVF00-0U, Eurobio), 0.1 mM non-essential amino acids (11140-035, Gibco), and 10 µg/mL gentamicin (15710-049, Gibco).

## Determination of the depth profile of the Varian Primus CLINAC

The depth profile of the Varian Primus CLINAC was established using a water tank and an ionization chamber. The Varian Primus CLINAC was used in photon mode at 20 MV. The measures were done at the isocenter.

## Irradiations of cells and mice

Cells were plated 10 hr before irradiation in 6-, 12-, or 96-well plates, according to the experiment. Plates were positioned straddling the light field boundary indicating the isodose 50 in order to get at least one well column inside the PTV and several well columns spreading over the margin. In each figure, the margin is indicated in terms of distance from the limit of the light field that is referred as the 0. At this position, the deposited dose is 50% of that in the PTV. The exact size of the margin varied with the used plate and its exact position in each experiment. Cells were irradiated using a Varian Primus CLINAC used in photon mode at 20 MV in all experiments, except in *Figure 3—figure supplement 4* where it was also used at 6 MV. In all experiments, nonirradiated plates were kept outside the bunker. Cells were always irradiated at 2 Gy, either once or each day except weekends, up to 3 weeks.

SKH1-E mice were anesthetized using isoflurane (5% for induction and 2% during the experiment with 0.8 L/min of oxygen). They were put beside a phantom made of PMMA plates. The contact between the mouse and the phantom was ensured by an ultrasound transmission gel. The irradiation was planned in depth so that 100% of the dose was deposited in the phantom at the level of the mouse flank skin and in width so that the isodose 50 was juxtaposed to the spine. A group of mice, used as positive control, was irradiated total body at 2 Gy. Another group of mice, used as negative control, was manipulated as the others and kept outside the bunker. The irradiations were performed using an Elekta Precise accelerator used in photon mode at 6 MV.

## Determination of the dose deposited in well culture plates by the Varian CLINAC

96-, 12-, or 6-well culture plates laid down on a 4-cm-PMMA plate were imaged using a tomodensi-tometer. A dosimetry study was then performed using the System Treatment Planning of the Centre Oscar Lambret for an irradiation beam of 20 MV covering half of the plate, the other half being in the margin.

## Cell death analysis

Vibrant FAM polycaspase assay (V35117, Molecular Probes) was used as recommended by the manu-facturer. Briefly, cells were harvested and incubated with the kit reagent solution for 1 hr at 37°C. After having been washed with the wash buffer provided in the kit, cells were incubated with propidium iodide (PI) (1 µg/mL) for 15 min on ice. Then, cells were analyzed by flow cytometry on a BD FACSCanto II (Becton Dickinson). Data analyses were made using the FACSDiva 6.0 software (BD Biosciences).

The Annexin V/PI assay (V13241, Molecular Probe) was used as recommended by the manufacturer. Briefly, cells were harvested, incubated with Alexa Fluor488-Annexin V and PI (1 µg/mL) for 15 min at room temperature. Then, cells were diluted in the buffer provided by the kit and analyzed by flow cytometry on a BD FACSCanto II (Becton Dickinson). Data analyses were made using the FACSDiva 6.0 software (BD Biosciences).

### SA-β-Gal assays

Cells were incubated for 2 hr with 33 µM $C_{12}$FDG (a fluorogenic substrate of β-galactosidase). Cells were washed with PBS and analyzed by flow cytometry for the forward scatter (FSC) and side scatter (SSC) values and $C_{12}$FDG intensity on a BD FACSCanto II cytometer (Becton Dickinson) or on a BD Influx (Becton Dickinson). Data analyses were made using the FACSDiva 6.0 software (BD Biosciences) or FlowJo V10.

Alternatively to using $C_{12}$FDG as a substrate for the β-galactosidase, we used X-Gal. Cells were fixed with 2% formaldehyde/0.2% glutaraldehyde in PBS for 4 min on ice. After a rapid wash with PBS, cells were incubated at 37°C for 7 hr in the X-Gal reaction mixture (1 mg/mL X-Gal, 40 mM phosphate buffer [pH 6], 5 mM potassium ferrocyanide, 5 mM potassium ferricyanide, 150 mM NaCl, 2 mM $MgCl_2$). The blue cells considered as SA-β-Gal positive were manually counted in at least five independent fields for a total of at least 100 cells for each condition.

### Cell cycle analysis

Cells were fixed overnight in ethanol 70% at 4°C. Then, they were rinsed with PBS and incubated 30 min at 37°C with PBS containing 10 µg/mL RNase A, 20 µg/mL PI, and 0,5% Triton X-100. Cells were kept on ice until analysis with a BD FACSCanto II cytometer (Becton Dickinson). Data analyses were made using FlowJo V10.

### Immunofluorescence

Cells grown in 96-well plates or on coverslides in 6- or 12-well plates were fixed on ice with cold acetone/methanol (v/v) or with PFA 4% at room temperature for 10 min and washed with PBS. Cells were permeabilized with a Triton X-100 solution at 0.2% in PBS followed by PBS washing. Unspecific sites were blocked by incubation at room temperature with a 5% non-fat milk solution in PBS. Then, epitopes were detected by incubation overnight at 4°C with antibodies against XRCC1 (ab47920 or ab1838, Abcam; sc-11429, Santa Cruz Biotechnology), 53BP1 (sc-22760, Santa Cruz Biotechnology), phosphoXRCC1 (A300-059A, Bethyl Laboratories), hOGG1 (B01P, Abnova), PNKP (Ab 170954, Abcam), or PAR chains (AM80, Calbiochem). After washing with PBS, cells were incubated with the secondary antibody (Life Technologies, A21206 and A-21202) followed by nuclear staining with 300 nM DAPI (Life Technologies, D1306). For the co-detection of the co-localization of 53BP1 and γH2AX, the antibodies against 53BP1 (NB100-304, Novus Biologicals) and γH2AX (NB100-78356, Novus Biologicals) were co-incubated overnight at 4°C. After washing with PBS, cells were co-incubated with the secondary antibodies (Life Technologies, A21206 and A-21202) followed by nuclear staining with 300 nM DAPI (Life Technologies, D1306).

The 96-well plates were analyzed on a high-content microscope (Operetta High-Content Imaging System). For each well, a hundred images were taken. The images were analyzed using the Columbus software (associated to the machine) to detect the nuclei and the fluorescent foci inside the nuclei. The coverslides were analyzed using a confocal microscope LSM 880 or an AxioImager Z1-Apotome from Zeiss. Images were analyzed using Zen (Zeiss) or the Biovoxel module of ImageJ (NIH), or for the detection of the co-localization of 53BP1 and gH2AX the JACoP module from Image J (NIH).

### Multinucleated cell counting

Immunofluorescence experiments were reanalyzed to count the number of nuclei per cells. Fluorescent microscopy photos were taken with the AxioImager Z1 Apotome (Zeiss) with the filter sets from Zeiss 43 (excitation: BP 550 ± 25; emission: BP 605 ± 70) or 38 (excitation: BP 470 ± 40; emission: BP 525 ± 50). Multinucleated cells were manually counted. We considered a cell as multinucleated when two or more nuclei were located in the same continuous fluorescent area marking the cytoplasm.

### Comet assays

Cells were suspended in low-melting point agarose at 0.5% in PBS at 42°C. The suspension was then immediately spread on a cometslide (4250-200-03, Trevigen). Agarose was allowed to cool down

20 min at 4°C. Then, cell membranes were permeabilized with a lysis solution (1.2 M NaCl, 100 mM EDTA, 10 mM Tris-HCl, 1% Triton X-100 [pH 10]) at 4°C, for 1 hr. Slides were then equilibrated in electrophoresis buffer (for pH 8: 89 mM Tris-base, 89 mM boric acid, and 2 mM EDTA; for pH 12.3: 300 mM NaOH, 1 mM EDTA) at 4°C. Then, an electrophoresis field of 40 V (15 mA) was applied during 30 min for pH 8 and of 60 V (90 mA) during 5 min for pH 12.3 at 4 °C. The electrophoretic migration was stopped by neutralizing the pH in a bath of cold water for 10 min. DNA was stained with SYBRGreen (X1000; Molecular Probes) according to the manufacturer's recommendation. The slides were photographed under an AxioImagerZ1 Apotome (Zeiss) microscope. The images were analyzed using an ImageJ in-home macro in which the head (the nucleus) and the tail (the DNA that migrated) of the comet are delimited in order to get the fluorescence intensity of the head, the fluorescence intensity of the tail, and the length of the tail. The calculation of tail moments was done using the formula: (length of the comet tail × fluorescence intensity of the tail)/total fluorescence intensity (head+ tail).

## Mouse skin histology

One hour after irradiation, mice were sacrificed and skin samples were dissected. They were fixed in 10% formalin solution, dehydrated, and paraffin-embedded. Immunohistofluorescence was performed on 6-µm-thick sections. Slides were deparaffinized and then antigen retrieval was made by incubation in a sub-boiling 10 mM sodium citrate buffer. Tissues were permeabilized in a 0.4% Triton X-100 solution, and nonspecific binding was blocked with a 5% BSA solution in PBS for 2 hr. Sections were then incubated overnight with anti-XRCC1 (ab134056 from Abcam) or anti-53BP1 (NB100-304, Novus Biologicals) antibodies. After washes, the secondary antibody (A10042) was incubated for 45 min at room temperature. After washes, the nuclei were stained by a 300 nM DAPI solution (Life Technologies, D1306). The slides were analyzed using a Zeiss LSM 880 confocal microscope. The number of foci per nuclear focal plan was determined using the speckle inspector function of the Biovoxel module of ImageJ.

For the co-detection of XRCC1 foci and vimentin, skin samples were frozen immediately after dissection in a cold isopentane solution. Immunohistofluorescence was performed on 5-µm-thick sections. Tissues were post-fixed with a solution of 4% PFA. Tissues were permeabilized in a 0.25% Triton X-100 solution, and nonspecific binding was blocked with a 5% BSA solution in PBS for 2 hr. Sections were then incubated overnight at 4°C with anti-XRCC1 (#2735S from Cell Signaling) and anti-vimentin (AF2105 from R&D) antibodies. After washes, the secondary antibodies (A10042 and A11055 from Molecular Probes) were incubated for 1 hr at room temperature. After washes, the nuclei were stained by a 300 nM DAPI solution (Life Technologies, D1306). The slides were analyzed using a Zeiss LSM 710 confocal microscope. The number of foci per nuclear focal plan of cells positive for vimentin was determined using the speckle inspector function of ImageJ.

## Senescent cell sorting

Senescent NHDFs were sorted on a BD Influx (Becton Dickinson) equipped with a 200 µm nozzle, tuned at a pressure of 3.7 psi and a frequency of 6200 kHz. Sample fluid pressure was adjusted to reach an event rate of 1000 events/s. Senescent cells were gated as those having the highest fluorescent intensity for the $C_{12}FDG$ staining accompanied by the highest FSC and SSC values. This senescent population was electrostatically sorted in air, collected in complete culture medium, and cultured again as described above.

## In-gel zymography assays

The presence of latent and active forms of MMPs secreted into the culture medium was assayed as previously described (*Lee et al., 2016*) Cells were incubated overnight with a serum-free (i.e., gelatinase-free) medium. Cell culture media were collected and concentrated using a Vivaspin 6.5 kDa (VS0611, Sartorius Stedim). Cell culture media were diluted in 4× nondenaturing sample buffer (0.5 M Tris-HCl pH 6.8, 40% glycerol, 10% SDS, 0.1% bromophenol blue) and separated on 10% acrylamide-bisacrylamide (v/v) gels containing 1% gelatin as an MMP substrate. After running at a steady voltage of 125 V for 90 min, the gels were incubated at room temperature with 2.5% Triton X-100 for 1 hr. Subsequently, the gels were incubated in 50 mM Tris-HCl pH 7.5, containing 0.2 M NaCl, 5 mM $CaCl_2$, and 0.25% Triton X-100 at 37°C for 15 hr. Then, they were stained for 3 hr with a

solution containing 0.5% (w/v) Coomassie Brilliant Blue R-250, 5% (v/v) methanol and 10% (v/v) acetic acid, then destained for getting contrast with 10% (v/v) methanol and 5% (v/v) acetic acid.

## Migration and invasion assays

NHDFs were suspended in a serum-free FGM-2 medium at $1 \times 10^5$ cells/mL. 200 µL of this suspension was loaded on the top of Boyden chambers, uncoated for migration assays (353097, Corning) or coated with Matrigel for invasion assays (354480, Corning). A complete FGM-2 medium with serum was added in the lower chamber. Cells were incubated for 24 hr at 37°C with 5% $CO_2$. Afterward, cells were fixed with cold ethanol on ice for 10 min. Cells remaining on the upper side of the filters were removed using a cotton swab. Cells that reached the bottom side of the filters were stained with a 0.25% crystal violet (C3886, Sigma-Aldrich) solution in 70% ethanol and manually counted.

## HPRT assays

Cells were exposed to 100 µM 6-TG (A4882, Sigma) twice per day for up to 14 days. At different time points, cells were stained with 0.05% crystal violet in 70% ethanol. For quantifying the results, the crystal violet was redissolved in 2% SDS in distilled water. The color intensity was then quantified by measuring the absorbance at 570 nm. To get the percentage of 6-TG-resistant cells, the absorbance value of 6-TG-treated cells was divided by that of DMSO-treated cells and normalized by that of cells at day 0.

## Tumorigenesis assay

$1 \times 10^6$ fibroblasts of each condition were suspended in 100 µL of FGM-2 bulletkit medium (CC-3132, Lonza) containing 50% type I collagen (A10483-01, Gibco). The suspensions were injected subcutaneously in the flank of CB17-SCID mice. For positive control, $1 \times 10^6$ HT1080 cells suspended in 100 µL of DMEM containing 50% type I collagen (A10483-01, Gibco) were also injected subcutaneously in the flank of CB17-SCID mice. 12 mice were injected with nonirradiated NHDFs, 22 mice with PSNE cells, and 10 mice with HT1080 cells.

## Quantitative real-time PCR

RNAs were extracted using the NucleoSpin RNA kit (REF 740955, Macherey-Nagel). 1 µg of total RNA was reverse-transcribed with Superscript IV (18091050, Invitrogen) completed with random hexamers and dNTPs according to the manufacturer's instruction. Mx3005P (Agilent, Santa Clara, CA) was used for the real-time PCR. Primers were designed with the qPrimerDepot software (http://primerdepot.nci.nih.gov/). For p16 (*cdkn2a*): TGCCTTTTCACTGTGTTGGA and GCCATTTGCTAGCAGTGTGA; for EAR: GAGGCTGAGGCAGGAGAATCG and GTCGCCCAGGCTGGAGTG; for p21 (*cdkn1a*): ATGA AATTCACCCCCTTTCC and CCCTAGGCTGTGCTCACTTC. PCR products were detected using SYBR Green fluorescence (SYBR Green Master Mix, Life Technologies). Measures were performed in triplicate for each data point. Results were analyzed by the manufacturer software MxPro (Agilent). The expressions of p16 and p21 were normalized to that of EAR.

## JC-1 staining

Mitochondrial viability was assessed by incubating cells with the JC-1 probe (T3168, Thermo Fisher) at 1 µM in FBM medium without serum and antibiotics for 30 min at 37°C. Then, NHDFs were harvested and washed twice in PBS. The fluorescence was analyzed with a BD FACSCanto II cytometer (Becton Dickinson) at 488 nm (green) and 590 nm (red). Data analyses were made using the FACSDiva 6.0 software (BD Biosciences). The ratio red/green was normalized on that of non-treated NHDFs.

## Western blots

Cells were directly lysed in Laemmli buffer (25 mM Tris-HCl pH 6.8; 2% SDS; 10% glycerol; 2.5% β-mercaptoethanol; 0.01% bromophenol blue). Proteins were separated by SDS-polyacrylamide gel electrophoresis and transferred on nitrocellulose membranes (88018, Thermo Fisher Scientific). Membranes were blocked with 5% non-fat dried milk or 5% BSA in PBS. Then, membranes were incubated overnight with the primary antibody at the dilution recommended by the manufacturer: anti-PAR (AM80, Calbiochem), anti-GAPDH (sc-32233, Santa Cruz Biotechnology), anti-Rb (#9309, Cell Signaling), anti-phospho-Rb (#9308, Cell Signaling), and anti-p16 (550834, BD Pharmingen).

Secondary antibodies used were anti-mouse and anti-rabbit peroxidase conjugated (respectively 715-035-151, 711-035-152, Jackson ImmunoResearch Laboratories) or anti-mouse fluorescent dye conjugated secondary antibody (926-32210, LI-COR). The peroxidase activity was revealed using an ECL kit (RPN2106, Amersham Biosciences) or SuperSignal West Dura Extended Duration Substrate (34076, Thermo Fisher Scientific). For fluorescent Western blots, the total proteins used for normalization were detected by using the Revert 700 nm Total Protein Stain (926-110.21, LI-COR).

## Statistical analyses

We used the D'Agostino–Pearson normality test to determine if the data distribution was normal. When the data distribution followed a normal law, we used one-way ANOVA to evaluate the differences among more than three groups and/or Student's $t$-test to evaluate the differences between two groups. When the data distribution was not normal, we used a Kruskal–Wallis test to evaluate the differences among more than three groups and/or a Wilcoxon test to evaluate the differences between two groups. We used a chi-squared test to evaluate the difference between populations of 10,000 cells analyzed by flow cytometry. The p-values are indicated in the diagrams as $*p<0.05$, $**p<0.01$, or $***p<0.001$. When $p>0.05$, differences were considered as nonsignificant and noted as NS.

## Acknowledgements

This work was supported by the Centre National de la Recherche Scientifique, the Université de Lille, the Ligue contre le Cancer (Comité du Pas-de-Calais, Comité de la Somme, Comité du Nord), the Cancéropole Nord-Ouest, the Institut Pasteur de Lille, the SIRIC OncoLille (Grant INCa-DGOS-Inserm 6041), the Agence Nationale de Recherche (ANR-10-EQPX-04-01), the Feder (12001407 [D-AL]), and the Contrat de Plan Etat Région CPER Cancer 2015-2020. EG had a fellowship from the Institut Pasteur de Lille and the Région Hauts-de-France. MT had a fellowship from the Université de Lille. CF had a fellowship from the European Erasmus program. JN had fellowships from the Université de Lille and from the Association pour la Recherche sur le Cancer. LS and CD had fellowships from the Région Hauts-de-France. We thank Thomas Lacornerie for giving access to the Varian CLINAC and for performing dose profiles. We thank the Bioimaging Center Lille-Nord de France (Campus Calmette), especially Antonino Bongiovanni and Hélène Bauderlique, for imaging and cytometry facilities. We thank the PLETHA animal facility, especially Thierry Chassat and David Hannebique. We thank David Dombrowicz for giving access to the BD Influx (Becton Dickinson). We thank Benoit Vatrinet, Anaïs Engrand, and Olivier Samyn for technical help. The authors have no conflicting financial interests.

## Additional information

### Funding

| Funder | Grant reference number | Author |
| --- | --- | --- |
| Ligue Contre le Cancer | | Corinne Abbadie |
| SIRIC Oncolille | INCa-DGOS-Inserm 6041 | Corinne Abbadie |
| Agence Nationale de la Recherche | ANR-10-EQPX-04-01 | Priscille M Brodin |
| Feder | 12001407 (D-AL) | Priscille M Brodin |
| Institut Pasteur de Lille, France | PhD student Fellowship | Erwan Goy |
| Region des Hauts-de-France, France | PhD student Fellowship | Erwan Goy |
| European Erasmus Program | Graduate student Fellowship | Caterina Facchin |
| Fondation ARC pour la Recherche sur le Cancer | PhD student Fellowship | Joe Nassour |

| Funder | Grant reference number | Author |
|---|---|---|
| Region des Hauts-de-France | Engineer Fellowship | Laure Saas |
| Region des Hauts-de-France | Post-doctoral Fellowship | Claire Drullion |

The funders had no role in study design, data collection and interpretation, or the decision to submit the work for publication.

## Author contributions

Erwan Goy, Caterina Facchin, Nathalie Martin, Emmanuel Bouchaert, Jerome Benoit, Joe Nassour, Laure Saas, Priscille M Brodin, Alexandre Vandeputte, Olivier Molendi-Coste, Laurent Pineau, Gautier Goormachtigh, Olivier Pluquet, Albin Pourtier, Fabrizio Cleri, Conceptualization, Data curation, Formal analysis, Funding acquisition, Methodology, Supervision, Writing – original draft; Maxime Tomezak, Conceptualization, Data curation, Formal analysis, Methodology, Writing – original draft; Clementine de Schutter, Investigation, Methodology; Claire Drullion, Methodology; Eric Lartigau, Corinne Abbadie, Conceptualization, Data curation, Formal analysis, Funding acquisition, Methodology, Resources, Supervision, Writing – original draft; Nicolas Penel, Conceptualization

## Author ORCIDs

Priscille M Brodin ⓘD http://orcid.org/0000-0003-0991-7344
Corinne Abbadie ⓘD http://orcid.org/0000-0002-8174-2393

## Ethics

This study has received a favorable statement from our ethic committee (CEEA 75. Nord Pas-de-Calais, France) and was authorized by the Ministere de l'education nationale, de l'enseignement superieur et de la recherche under the number APAFIS#13531-2018070414101586v1.

## Decision letter and Author response

Decision letter https://doi.org/10.7554/eLife.67190.sa1
Author response https://doi.org/10.7554/eLife.67190.sa2

# Additional files

## Supplementary files

• Transparent reporting form

## Data availability

All data generated or analyzed during this study are included in the manuscript and supporting files. Source data files have been provided for all the figures where it was needed.

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
