## [Editor Report]

A major issue from radiation therapy is the generation of secondary tumors that can arise a long time after treatment. Here, the authors use careful and innovative experimental systems to show that out-of-field dose scattered radiation induces a senescent arrest characterized by single-strand DNA breaks (SSB) and the ability to escape from the senescent arrest albeit at a very low frequency. The data are consistent with a key role for SSBs and the ability of the cells to escape from senescence, and the paper has a clear clinical potential: the prevention of secondary tumors using senolytic agents.

---

## [Decision Letter]

**Decision letter after peer review:**

Thank you for submitting your article "The out-of-field dose in radiation therapy induces delayed tumorigenesis by senescence evasion" for consideration by *eLife*. Your article has been reviewed by 2 peer reviewers, and the evaluation has been overseen by a Reviewing Editor and Wafik El-Deiry as the Senior Editor. The reviewers have opted to remain anonymous.

Essential revisions:

1) Please address the question of sensitivity for DSB as measured by the comet assay. An independent biomarker of DSBs, such as γH2AX, has been suggested.

2) Please provide representative images of the foci for Figure 3A.

3) Please address the question related to the use of NAC to impact DSB production (Reviewer 2, Page 11 Lines 224-227 and Lines 248-252).

4) Please address all typographical errors identified by the reviewers.

*Reviewer #1:*

This is an original contribution based on previous work from the Abbadie group describing the emergence of tumorigenic cells from senescent cells. The authors use careful and innovative experimental systems to show that out of field dose scattered radiation induce a senescent arrest characterized by single stranded DNA breaks (SSB) and the ability to escape form the senescent arrest albeit at a very low frequency. The main conclusion although interesting is still correlative. The authors do not show whether SSBs are sufficient or required for the emergence of pro-tumorigenic cells from senescent cells. The data presented is nevertheless consistent with a key role for SSBs with the ability of the cells to escape from senescence. Beyond the mechanistic aspects, this paper has a clear clinical potential: the prevention of secondary tumors using senolytic agents.

The following remarks need authors attention and clarification:

1. In figure 2, senescence is characterized by using the classic SA-b-Gal marker and cell morphology parameters obtained by FACS. This paper aims to distinguish senescence with SSBs and DSB as two important types of senescent cells. A more detailed characterization is needed to fully understand these differences. In particular the SASP is a very important trait of senescence linked to the ability of senescent cells to stimulate tumorigenesis. This should ideally be characterized.

2. In figure 3A, the number of DSB observed in the PTV according to the p53BP1 biomarker is three times or more in comparison with cells in the margin. However, in figure 3B there is just a minor difference. This suggests that the sensitivity of the comet assay used is very low and cannot be used to conclude "a lack" of DSBs in cells in the margin. An independent biomarker of DSBs such as γH2AX is suggested to reinforce this important point.

3. Figure 3A should also include representative images of the foci quantified by the authors. Low resolution images are provided in figure 4B, but no foci can be distinguished in those pictures. Since the authors quantify foci to support their conclusions they need to be shown.

4. What is the identity of the dermal cells showing SSBs in figure 4B. The paper focuses on the production of sarcomas by senescent fibroblasts. Ideally, the cells showing SSBs after irradiation should be fibroblasts so the in vivo data can support the data obtained in vitro.

5. Lane 316 the phrase: "were ever fully repaired" is not clear, why use the word ever here?

6. The decline in the ability of making PAR chains is a very interesting observation. The authors propose that this is due to NAD depletion. However, low PARP levels (this was discarded in the discussion) or high PARG activity can equally explain these results. A direct measurement of NAD levels will be very informative.

7. The malignant properties of the fibroblasts that emerge from senescence are not fully characterized. The ability of these cell to form tumors in vivo would ideally be provided to conclude that these cells are responsible for secondary cancers after radiation.

8. The idea that SSB lead to a senescence arrest from which cells can escape is interesting and have wide ranging biological consequences. The authors end their paper showing that treating cells with reagents that induce only SSBs are sufficient to trigger this phenomenon. However, cells that are irradiated in the PTV also show high levels of SSB and they do not generate cells that escape from senescence. There are two competing models that explain the data. First, the author's model: "SSB accumulation is sufficient to induce a senescence stage form which cells can escape". Alternatively, the senescence arrest associated to SSBs is not complete and the SSBs are just a measure of lower level of damage but not mechanistically involved in generating post-senescence cells. In fact, the emergence of post senescence cells is a very rare phenomenon and that is not consistent with the use of the word sufficient to explain the role of SSBs.

9. It would be relatively easy for the authors to show that senescent cells in the margin having mostly SSBs are sensitive to senolytics. Can these data be provided?

*Reviewer #2:*

One major issue from radiation therapy is the generation of secondary tumors that can arise a long time after treatment. This raises issues both to the academic point of view, to know cellular and molecular events involved, and for medical treatment. In this work, the authors designed a precise and elaborate protocol mimicking exposure of adjacent cells (margin ) to the central radiation beam, i.e. the planning target volume (PTV). The authors deeply characterized this novel irradiation protocol.

DNA double strand breaks (DSB) and single strand breaks (SSB) are the main DNA breaks generated by ionizing radiation (IR), but DSBs are generally considered as the most harmful lesion, leading to cell death. The authors made several observations, many of them being unexpected:

– Both DSB and SSB are produced in the PTV, while only (or mostly) SSB are generated in the margin. This leads to mainly cell death in the PTV, but mainly senescence in the margin.

– Senescence in the margin result from SSB accumulation due to SSB repair defect.

– Some cells escape from senescence, bearing mutations and markers of transformed cells.

This allows to establish a model with all steps elucidated.

Comments for the authors:

1. Page 8, line 133: "normal human dermal fibroblasts": Please confirm that these are primary dermal fibroblasts.

2. Page 9, line 169-175, Figure 1C and Supp: sounds there are a little bit more caspase-independent death à 4x2Gy with F6MC1. Is it linked to the age of the donor or to individual variability? Also there is late apoptosis at 6x2Gy, no? Could the authors comment?

3. Page 9 line 187 (Figure 2A). Seems there is no decrease in the margin.

4. Page 11 Line 224-227: did the authors tried to analyze the impact of NAC, on DSB production? For some authors the DSB indirectly arise from bases oxidation (resulting from water radiolysis) and incision of the oxidized bases on the complementary DNA strands. According to this hypothesis, reducing ROS by treating cells with NAC might reduce the frequency of DSBs. Same comment lines 248-252.

5. Page 12 line 268. According to the cell cycle analysis, many cells are not so much quiescent, the effects are moderate and many cells are still replicating their genome. Did the authors try without any serum (primary cells can support this) or at confluency? And also at low O2 concentration (for example, 3%) that better mimics the in vivo conditions.

6. Page 13 line 305: "SSBs were less numerous after five irradiations than after just one". Could the authors comment (one sentence might be sufficient).

7. Page 13 line 310: please define SSBR in the text (I suppose it is SSB repair).

8. Page 13 Line 310-319: can we envision the de novo production of SSB production, that might alter the final quantification?

9. The discussion is a little bit too long. For instance the paragraph on the senolytics (line 461-480) could be shortened. It is not necessary to deeply detail all the potential strategies; this is interesting, but a little bit out the scope of the paper.

---

## [Author Response]

Reviewer #1:This is an original contribution based on previous work from the Abbadie group describing the emergence of tumorigenic cells from senescent cells. The authors use careful and innovative experimental systems to show that out of field dose scattered radiation induce a senescent arrest characterized by single stranded DNA breaks (SSB) and the ability to escape form the senescent arrest albeit at a very low frequency. The main conclusion although interesting is still correlative. The authors do not show whether SSBs are sufficient or required for the emergence of pro-tumorigenic cells from senescent cells. The data presented is nevertheless consistent with a key role for SSBs with the ability of the cells to escape from senescence. Beyond the mechanistic aspects, this paper has a clear clinical potential: the prevention of secondary tumors using senolytic agents.

We agree with this summary, except regarding the sentence “the authors do not show whether SSBs are sufficient or required for the emergence…”. The results presented in Figure 6 (completed by several figure supplements) do show that SSBs are sufficient for inducing senescence followed by post-senescence emergence when they are generated in the absence of DSBs and other cell damages. See also answer to point 8.

The following remarks need authors attention and clarification:1. In figure 2, senescence is characterized by using the classic SA-b-Gal marker and cell morphology parameters obtained by FACS. This paper aims to distinguish senescence with SSBs and DSB as two important types of senescent cells. A more detailed characterization is needed to fully understand these differences. In particular the SASP is a very important trait of senescence linked to the ability of senescent cells to stimulate tumorigenesis. This should ideally be characterized.

Although the question of whether the SASP could differ between SSB- or DSB-induced senescence, and whether these two potential types of SASP could differently stimulate tumor development is very interesting, but this was not the purpose of this study.

2. In figure 3A, the number of DSB observed in the PTV according to the p53BP1 biomarker is three times or more in comparison with cells in the margin. However, in figure 3B there is just a minor difference. This suggests that the sensitivity of the comet assay used is very low and cannot be used to conclude "a lack" of DSBs in cells in the margin. An independent biomarker of DSBs such as γH2AX is suggested to reinforce this important point.

it is established that amongst the DNA breaks induced by ionizing radiations, SSBs are much more numerous than DSBs, what our results confirm. DSBs are more easily quantifiable than SSBs by immunofluorescence, because the 53BP1 foci are bigger than the XRCC1 foci (see the new Figure 3—figure supplement 1). In contrast, SSBs are more easily quantifiable than DSBs by comet assay because the comet tails are longer. Therefore, the two techniques are complementary. In figure 3A, the differences in DSB numbers between the PTV and the margin are indeed at least 3-fold. In figure 3B, they are of the same range, between 2.1 and 2.5 (see the exact values in Author response table 1), but this is not very easy to see on the graph because of the scale we have to put to include all the extreme values.

**Author response table 1. sa2table1:** 

	Non-irradiated	PTV	Margin (-5 to +20mm)	Margin (+22 to +47mm)
Mean	2.34	7.49	2.96	3.45
Fold change between PTV and margin			2.53	2.17

However, we have done immunofluorescence against γH2AX as requested. The results are given in a new Figure 3—figure supplement 2. We have quantified 53BP1 foci, γH2AX foci and the double-stained foci. Whatever the marker used, the results are similar and confirm the absence or quasi absence of DSBs in cells positioned in the margin, compared to those positioned inside the PTV.

3. Figure 3A should also include representative images of the foci quantified by the authors. Low resolution images are provided in figure 4B, but no foci can be distinguished in those pictures. Since the authors quantify foci to support their conclusions they need to be shown.

We have now given representative images of 53BP1 and XRCC1 foci in Figure 3—figure supplement 1. We have also done again the images of figure 4B at a better resolution.

4. What is the identity of the dermal cells showing SSBs in figure 4B. The paper focuses on the production of sarcomas by senescent fibroblasts. Ideally, the cells showing SSBs after irradiation should be fibroblasts so the in vivo data can support the data obtained in vitro.

We have now performed a co-detection of XRCC1 foci with vimentin as a marker of fibroblasts, which appear as the more numerous cells in this tissue. We have quantified the XRCC1 foci in the vimentin-positive cells. The results are not significantly different from the quantification of XRCC1 foci in unidentified dermal cells. We added these results in a new Figure 4—figure supplement 1.

5. Lane 316 the phrase: "were ever fully repaired" is not clear, why use the word ever here?

It is a mistake. We wanted to say “always”. This is corrected in the new version.

6. The decline in the ability of making PAR chains is a very interesting observation. The authors propose that this is due to NAD depletion. However, low PARP levels (this was discarded in the discussion) or high PARG activity can equally explain these results. A direct measurement of NAD levels will be very informative.

We have made the same hypotheses as the reviewer, and these points are currently under investigation, but none of our results are presently convincing. Our latest results suggest that the expression of PARP1 could decline a bit in contrast to what we wrote in the initial version of the paper. So, we have changed the text, to just say that the mechanisms are presently unknown and under investigation.

7. The malignant properties of the fibroblasts that emerge from senescence are not fully characterized. The ability of these cell to form tumors in vivo would ideally be provided to conclude that these cells are responsible for secondary cancers after radiation.

We have done this experiment, but at the time we submitted our article, we though that the delay of observation was too short to conclude. Now, we are 10 months after grafting, so we give the results as they are (see p15).

8. The idea that SSB lead to a senescence arrest from which cells can escape is interesting and have wide ranging biological consequences. The authors end their paper showing that treating cells with reagents that induce only SSBs are sufficient to trigger this phenomenon. However, cells that are irradiated in the PTV also show high levels of SSB and they do not generate cells that escape from senescence. There are two competing models that explain the data. First, the author's model: "SSB accumulation is sufficient to induce a senescence stage form which cells can escape". Alternatively, the senescence arrest associated to SSBs is not complete and the SSBs are just a measure of lower level of damage but not mechanistically involved in generating post-senescence cells. In fact, the emergence of post senescence cells is a very rare phenomenon and that is not consistent with the use of the word sufficient to explain the role of SSBs.

This comment of the reviewer made me realize that I was not enough clear. What is important for senescence escape is the accumulation of unrepaired SSBs in the absence of DSBs, shortened telomeres or other DNA damages able to activate the DDR pathway. I now better explain that in the discussion. Moreover, I used in the new version of the article, wherever possible, the expressions “SSB-without-DSB signature” or “SSB only-induced senescence” instead of “SSB-induced senescence”.

For us, the fact that only a few cells escape the senescent arrest do not preclude the use of the word “sufficient”. We used the word “sufficient” in its mathematical acceptation meaning that if SSBs (without DSBs) are present, senescence and post-senescence emergence will occur. The fact that only a few cells escape senescence could be explained by the mutations induced by the SSB that could only stochastically affect genes involved in cell cycle regulation or other key functions involved in transformation. However, we agree that we have not yet investigated how SSB mechanistically induce senescence and escape. I added a sentence for saying that in the revised version.

9. It would be relatively easy for the authors to show that senescent cells in the margin having mostly SSBs are sensitive to senolytics. Can these data be provided?

We have done these experiments and showed that Navitoclax was indeed able to kill senescent fibroblasts arising in the PTV or in the margin. Navitoclax was also able to kill MDA-MB 231 cells having resisted to 2 weeks of irradiations. These results are therefore highly promising for a future use in clinics. Presently, we are performing experiments in mice. We think it will be more pertinent to publish these results with those in vivo to come.

We also tried the association Dasatinib+Quercetin. These drugs were indeed toxic for senescent fibroblasts, but killed at the same level the proliferating ones.

Reviewer #2:Comments for the authors:1. Page 8, line 133: "normal human dermal fibroblasts": Please confirm that these are primary dermal fibroblasts.

These are indeed primary cells. I slightly changed the text to be clearer.

2. Page 9, line 169-175, Figure 1C and Supp: sounds there are a little bit more caspase-independent death à 4x2Gy with F6MC1. Is it linked to the age of the donor or to individual variability? Also there is late apoptosis at 6x2Gy, no? Could the authors comment?

In all these graphs, we gave the statistical analyses for the sum of all kinds of cell death, but we also performed the statistical analyses for each type of cell death. When it was non-significant globally, it was also non-significant for each type of cell death.

3. Page 9 line 187 (Figure 2A). Seems there is no decrease in the margin.

I do not understand the point.

4. Page 11 Line 224-227: did the authors tried to analyze the impact of NAC, on DSB production? For some authors the DSB indirectly arise from bases oxidation (resulting from water radiolysis) and incision of the oxidized bases on the complementary DNA strands. According to this hypothesis, reducing ROS by treating cells with NAC might reduce the frequency of DSBs. Same comment lines 248-252.

We have done experiments with NAC because we believed that ROS produced by water radiolysis would be at the origin of SSBs (rather than DSBs which are generally assumed to result from a direct ionization of DNA). Author response image 1 shows the growth curves of MDA-MB231 and NHDFs irradiated during two weeks in the presence or not of NAC. The results indicate that NAC does not rescue the growth of both cell types, whether they were irradiated or not, localized in the PTV or in the margin, rather NAC slightly slowed down the growth of NHDFs. This indicates that in this experimental setting, the impact of radio-induced oxidative stress is non-significant. We do not wish to publish these results, because they are preliminary and the role of ROS needs to be in-depth investigated (ROS concentration measures, other antioxidants).

**Author response image 1. sa2fig1:** 

5. Page 12 line 268. According to the cell cycle analysis, many cells are not so much quiescent, the effects are moderate and many cells are still replicating their genome. Did the authors try without any serum (primary cells can support this) or at confluency? And also at low O2 concentration (for example, 3%) that better mimics the in vivo conditions.

We tried to completely suppress the serum, FGF and insulin, but this was toxic for the NHDFs. Nevertheless, at 0.1% of serum, S phase cells are two-fold less, whereas DNA damages are the same. So we think our conclusions are valid.

– Putting the cells at confluency is not possible in studies regarding senescence, because confluent cells become SA-b-Gal-positive.

– 3% of O2 is not physiologic for skin cells which are mainly supplied by external oxygen (see for example Stucket et al., J Physiol, 2002, 538, 985-994; Evans et al., J Invest Dermatol, 2006, 126, 2596).

6. Page 13 line 305: "SSBs were less numerous after five irradiations than after just one". Could the authors comment (one sentence might be sufficient).

The reviewer points a result we do not really understand. Our sole hypothesis is that cells inside the PTV become more and more able to repair SSBs along irradiations. However, since this result concerned the PTV that was not the focus of this study, we did not further investigate this point, and we prefer not to comment it in the paper to avoid confusions.

7. Page 13 line 310: please define SSBR in the text (I suppose it is SSB repair).

SSBR was defined in the introduction section, lane 80

8. Page 13 Line 310-319: can we envision the de novo production of SSB production, that might alter the final quantification?

I am not sure to really understand the point. Does the reviewer speak about a de novo production of SSBs during the 72hrs following the last irradiation? If yes, this cannot be excluded, but this de novo production will take place only after 5 irradiations but not after just one. We think that it is unlikely.

9. The discussion is a little bit too long. For instance the paragraph on the senolytics (line 461-480) could be shortened. It is not necessary to deeply detail all the potential strategies; this is interesting, but a little bit out the scope of the paper.

We shortened this part of the discussion.